# TEXT2GRAD: REINFORCEMENT LEARNING FROM NATURAL LANGUAGE FEEDBACK

**Hanyang Wang**[1*]    **Lu Wang**[2†]    **Chaoyun Zhang**[2]    **Tianjun Mao**[3]
**Si Qin**[2]    **Qingwei Lin**[2]    **Saravan Rajmohan**[2]    **Dongmei Zhang**[2]
[1]University of Chicago    [2]Microsoft    [3]Fudan University

## ABSTRACT

Traditional RLHF optimizes language models with coarse, scalar rewards that mask the fine-grained reasons behind success or failure, leading to slow, opaque learning. Recent work augments RL with textual critiques through prompting or reflection, improving interpretability but leaving model parameters untouched. We introduce TEXT2GRAD, a reinforcement-learning paradigm that *turns free-form textual feedback into span-level gradients*. Given human (or programmatic) critiques, TEXT2GRAD aligns each feedback phrase with the relevant token spans, converts these alignments into differentiable reward signals, and performs gradient updates that directly refine the offending portions of the model's policy. This yields precise, feedback-conditioned adjustments instead of global nudges. TEXT2GRAD is realized through three components: (1) a high-quality feedback–annotation pipeline that pairs critiques with token spans; (2) a fine-grained reward model that predicts span-level reward on answers while generating explanatory critiques; and (3) a span-level policy optimizer that back-propagates *natural-language gradients*. Across summarization, code generation, and question answering, TEXT2GRAD consistently surpasses scalar-reward RL and prompt-only baselines, providing both higher task metrics and richer interpretability. Our results suggest that natural-language feedback can serve not only as explanations, but also as actionable training signals for fine-grained alignment. The code for our method is available at *https://github.com/microsoft/Text2Grad*.

## 1 INTRODUCTION

Free-form natural language feedback is abundant in real-world applications (Zhang et al., 2024a). Users leave suggestions in reviews, developers comment on code pull requests, and customers critique responses from virtual assistants. Unlike scalar ratings or preference scores, this form of feedback is inherently rich and expressive. It not only pinpoints what is correct or incorrect in an output but also explains why, providing actionable guidance for improvement.

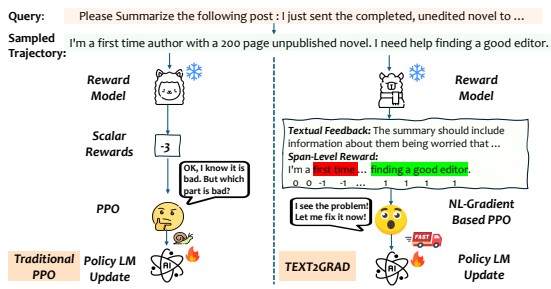

Figure 1: Comparison of PPO and TEXT2GRAD

Despite its ubiquity and usefulness, most learning paradigms fail to fully leverage human feedback. Reinforcement learning from human feedback (RLHF) has become the dominant method for aligning large language models (LLMs) with human preferences (Stiennon et al., 2020; Ouyang et al., 2022; Bai et al., 2022; Rafailov et al., 2023; Shao et al., 2024). RLHF typically reduces preference comparisons to scalar rewards and optimizes policies via PPO (Schulman et al., 2017) or DPO (Rafailov et al., 2023). While effective for improving helpfulness and safety, this scalarization discards fine-grained, token-level signals about what was right or wrong—and where—leading to

---

*First author. Email: `hanyangw@uchicago.edu`.
†Corresponding author. Email: `wlu@microsoft.com`.

imprecise credit assignment, slower convergence, and reduced interpretability (Casper et al., 2023; Wu et al., 2023; Raschka, 2024).

An alternative line of research maintains feedback in its natural language form. Methods such as ReAct (Yao et al., 2023) and Reflexion (Shinn et al., 2023) prompt the model to reflect on its outputs (Zhang et al., 2024b), generate critiques, and use them to self-correct in subsequent steps (Zhang et al., 2025). These approaches are inspired by how humans operate in open-ended tasks through reasoning, explanation, and dialogue, rather than relying on numerical rewards (Wei et al., 2022; Nakano et al., 2021; Zhang et al., 2024c). Natural language feedback in this context improves transparency and sometimes leads to better task performance. However, because these methods leave model parameters frozen, the feedback is not internalized, requiring repeated corrections and rendering it ephemeral (Fernandes et al., 2023; Pan et al., 2024; Sharma et al., 2024).

In this paper, we propose TEXT2GRAD, a novel framework that transforms free-form textual feedback into actionable gradients for policy optimization. As shown in Figure 1, unlike prior work that compresses feedback into scalar rewards or applies textual critiques only at inference time, TEXT2GRAD brings feedback into the training loop. Given a human or programmatic critique, our method aligns feedback clauses with relevant output token spans, converts these alignments into span-level reward signals, and computes a natural language gradient. This gradient is then used to perform policy updates that precisely adjust the parts of the model responsible for the error. The result is more targeted, efficient, and interpretable learning.

TEXT2GRAD is built on a complete pipeline for learning from text. First, we construct a high-quality annotation pipeline that uses GPT-4o to label model outputs with textual feedback and span-level critiques, following recent work on automated feedback generation (Lee et al., 2023; Liang et al., 2024). Second, we train a unified reward model inspired by generative reward modeling (Mahan et al., 2024) that jointly generates natural language critiques and structured span-level reward maps in a single autoregressive sequence. Third, we apply span-level policy optimization using a variant of PPO that integrates these fine-grained reward signals, drawing on advances in token-aware credit assignment (Chen et al., 2024a) and text-based gradients (Yuksekgonul et al., 2024).

We evaluate TEXT2GRAD across various domains, including summarization (Scheurer et al., 2023), code generation (Xu et al., 2025), and open-domain question answering (Cui et al., 2023). Our results demonstrate that by converting language into gradients, TEXT2GRAD not only achieves superior performance but also offers improved interpretability and sample efficiency, establishing natural language feedback as a powerful direct training signal. These results suggest that natural language feedback can be more than an interpretability tool: It can be converted into principled gradients to train more capable and aligned models. Overall, this paper makes the following contributions.

- We formulate the problem of learning from natural language feedback via gradient-based optimization, and present TEXT2GRAD as the first complete framework to address it.
- We develop a scalable annotation pipeline and a unified reward model that together produce span-level rewards and explanatory critiques, yielding *interpretable, span-level supervision*.
- We show that TEXT2GRAD outperforms strong scalar-reward-based and prompt-based baselines in summarization, code generation, and question-answering benchmarks.

TEXT2GRAD demonstrates that natural language feedback, when properly aligned and grounded, can serve as a direct training signal rather than just auxiliary guidance, opening a new path for building language models that learn from human-like supervision.

## 2 RELATED WORK

**RLHF with scalar rewards**  Reinforcement learning from human feedback replaces supervised labels with a reward model trained on pairwise human preferences (Christiano et al., 2017; Ouyang et al., 2022). The reward is a single scalar, and policy optimization methods such as PPO and DPO update the language model toward higher scores (Schulman et al., 2017; Rafailov et al., 2023). This recipe has advanced instruction following, safety, and summarization; a 1.3B InstructGPT model aligned in this way outperformed 175B GPT 3 on adherence and toxicity (Ouyang et al., 2022; Bai et al., 2022; Stiennon et al., 2020). Subsequent work studies reward hacking and data noise (Wang et al., 2024; Lambert, 2025; Sun et al., 2023). Despite these successes, scalar rewards collapse

multidimensional critiques into a single number, obscure the location of an error, and necessitate careful regularization, such as Kullback-Leibler penalties, to remain stable (Raschka, 2024; Wu et al., 2023). Even Process Reward Models (PRMs) (Lightman et al., 2023), which offer finer credit assignment, still rely on scalar signals and lack the explanatory power of natural language feedback. Recent work addresses credit assignment through span-level optimization: MA-RLHF improves efficiency via macro-action abstraction (Chai et al., 2024), SCAR decomposes scalar rewards using Shapley-value allocation (Cao et al., 2025), and Beyond Sparse Rewards generates intermediate numeric rewards via auxiliary language models (Cao et al., 2024).

**Natural language feedback at inference time** A complementary line of research keeps feedback in natural language but applies it only while the model is running. ReAct interleaves the chain of thought reasoning with tool use to refine answers in question answering and text games (Yao et al., 2023). Reflexion stores self-generated critiques between attempts and improves coding and decision tasks (Shinn et al., 2023). Language Feedback Training incorporates human-written refinements during supervised fine-tuning (Ouyang et al., 2022). Surveys categorize the many emerging feedback formats (Fernandes et al., 2023; Liang et al., 2024). These methods lift interpretability and sometimes quality, yet the model weights stay frozen, so lessons are not retained and error corrections must be rediscovered each time (Pan et al., 2024; Sharma et al., 2024). Learning from Natural Language Feedback (Chen et al., 2024b) uses free-form feedback to produce refined sequences and then imitation learns on those refinements under a supervised/KL view; the learning signal remains sequence-level targets derived from edited outputs.

TEXT2GRAD draws inspiration from both threads, yet differs crucially, by training a reward model that generates interpretable textual critiques, uniquely leveraging *natural language gradients* in token-level PPO to drive fast, interpretable policy improvements.

# 3 METHOD

This section details TEXT2GRAD, a novel framework for Reinforcement Learning from Natural Language Feedback. We define the Natural Language Gradient (NL-Gradient) and then describe our three-stage pipeline: (1) dual-feedback annotation, (2) generative reward modeling, and (3) NL-Gradient policy optimization that enables fine-grained learning from textual critiques.

## 3.1 NATURAL LANGUAGE GRADIENT: DEFINITION AND MOTIVATION

Traditional policy gradient methods optimize an expected scalar return $J(\theta) = \mathbb{E}_{y \sim \pi_\theta(\cdot|x)}\big[\mathcal{R}(y)\big]$, where $\mathcal{R}(y)$ is a sequence-level reward, which masks token-level contributions and hinders interpretability. To address this, we introduce the **NL-Gradient**, which transforms textual critiques into token-level gradient signals.

**Definition 1 (Natural Language Gradient)** *Given a generated sequence $y = (y_1, \ldots, y_T)$ and its textual critique $c$, let $\{\delta_t\}_{t=1}^{T}$ be token-level pseudo-rewards derived by aligning $c$ to $y$. The NL-Gradient is defined as:*

$$\nabla_{\text{NL}}(c \rightarrow y) \;=\; \sum_{t=1}^{T} \delta_t \, \nabla_\theta \log \pi_\theta(y_t \,|\, x, y_{<t}).$$

Note: "NL-Gradient" refers to converting language feedback into gradient-based supervision, not literal differentiation through text. We align critiques to spans, map spans to discrete token-level pseudo-rewards, and use these to weight the standard policy gradient. Natural language conditions *what* gets updated and *where*. Here, $\delta_t$ encodes the critique's local intensity on token $y_t$, enabling: (1) **Fine-Grained Guidance:** Pseudo-rewards $\delta_t$ highlight specific tokens needing improvement. (2) **Interpretability:** Each update step is grounded in human-readable feedback. (3) **Transferability:** The model learns a mapping from text to gradient signals, facilitating generalization between tasks. Our approach is compatible with both RLAIF and RLHF paradigms; human feedback experiments (Appendix I.1) demonstrate direct applicability to real human critiques.

## 3.2 OVERVIEW OF TEXT2GRAD

The core objective of TEXT2GRAD is to construct an NL-Gradient that directly drives policy updates. This requires solving two key challenges: (1) translating free-form textual critiques into structured

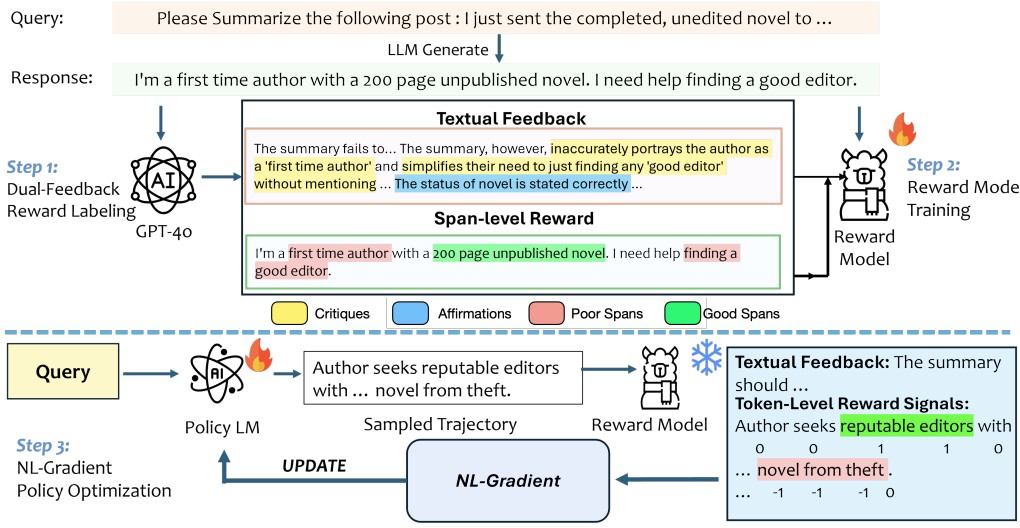

Figure 2: An overview of TEXT2GRAD. Yellow highlights critique phrases pointing out errors; Blue highlights affirming phrases identifying correct aspects.

token-level numerical feedback, and (2) leveraging these numerical signals to compute token-level advantages and update the policy. This establishes a principled bridge from linguistic reasoning to differentiable credit assignment, operating at a fundamentally different granularity than scalar RLHF methods. The framework generalizes across tasks without modification, requiring only a token-weighting wrapper on top of PPO. To address these challenges, as shown in Figure 2, TEXT2GRAD comprises three steps: *Dual-Feedback Reward Annotation*, which uses GPT-4o to produce high-quality paired critiques and scores; *Reward Model Training*, which trains a unified model to jointly produce explanatory critiques and structured span-level reward maps; and *NL-Gradient Policy Optimization*, which leverages per-token advantages and applies NL-Gradient PPO updates. Together, these phases realize end-to-end NL-Gradient descent for LLMs.

## 3.3 REWARD LABELING

Effective NL-Gradient optimization requires dense, interpretable feedback that can be precisely mapped to token-level learning signals. We introduce a **dual-feedback annotation framework** that jointly generates (1) free-form natural language critiques and (2) structured span-level reward labels. This design enables task-agnostic supervision while directly supporting the construction of token-level pseudo-rewards for fine-grained policy updates.

**Dual-Feedback Annotation** Given a prompt $x$ and a generated response $y = (y_1, \ldots, y_T)$, we aim to annotate each sample with a natural language critique $c$, describing strengths or weaknesses of the response in free text, and a structured span-level reward map $\mathcal{A}(y)$, where each span is assigned a label from $\{\texttt{positive}, \texttt{neutral}, \texttt{negative}\}$.

In practice, we prompt a strong LLM (e.g., GPT-4o) to output both feedback modalities. For example, in a summarization task, the model may generate a textual critique such as: *"The summary omits key information about the character's concern that the manuscript may be rejected."* followed by a structured JSON object assigning sentiment values to spans in the summary:

```
{
  "Good spans": ["200 page unpublished novel"],
  "Poor spans": ["first time author","finding a good editor"]
}
```

Critically, our annotation prompt explicitly requires spans to be grounded in and directly supported by the critique, ensuring semantic alignment. We annotate only `positive`/`negative` spans — the most informative signals — leaving `neutral` implicit, reducing overhead without loss of utility.

**Reasoning-Augmented Annotation** In the absence of human feedback, we employ **Chain-of-Thought (CoT) prompting** to elicit high-fidelity, self-justified annotations from GPT-4o. Given

a response $y$, the model: (1) Performs step-by-step quality reasoning; (2) Produces a critique $c$ grounded in that reasoning; (3) Derives a span-level reward map $\mathcal{A}(y) : s_k \mapsto \ell_k$, where each labeled span $s_k$ with label $\ell_k \in \{\texttt{positive}, \texttt{negative}\}$ must be explicitly anchored to evidence in $c$.

Formally, the reward labeler outputs: $R_{\text{LLM}}(x, y) = (c, \mathcal{A}(y))$, where $\mathcal{A}(y) : s_k \mapsto \ell_k$ maps each span $s_k$ to a label $\ell_k \in \{\texttt{positive}, \texttt{negative}\}$, explicitly justified by the critique $c$. This protocol enforces strict alignment between critique and annotation. Spans are labeled only where supported by prior reasoning, yielding semantically grounded, interpretable supervision without human references. Full prompts are provided in Appendix B.

**Token-Level Reward Mapping**  Although feedback is annotated at the span level, policy optimization requires token-level rewards. We convert each labeled span $s_k$ into token-aligned supervision by assigning a uniform pseudo-reward $\delta_t \in \{-1, 0, +1\}$ to each token:

$$\delta_t = \begin{cases} +1, & \text{if } t \in s_k \text{ and } \mathcal{A}(y)[s_k] = \texttt{positive}, \\ -1, & \text{if } t \in s_k \text{ and } \mathcal{A}(y)[s_k] = \texttt{negative}, \\ 0, & \text{otherwise.} \end{cases}$$

To reduce labeling cost while retaining informativeness, we adopt a class-prioritized strategy: only $\texttt{positive}$ and $\texttt{negative}$ spans are explicitly labeled, while $\texttt{neutral}$ spans are left unannotated and default to $\delta_t = 0$. This yields a token-level reward vector $\boldsymbol{\delta} = (\delta_1, \ldots, \delta_T)$, which supports token-wise advantage estimation and construction of the NL-Gradient (see Section 3.5). Our method does not impose fixed span lengths; spans are generated dynamically based on response content. Analysis on SLF5K (Tables 18 and 19) shows that performance depends on span selection quality rather than coverage: CoT-guided annotation labels 30% of tokens with precise signals while maintaining 93–96% accuracy across all span lengths, whereas dense per-token labeling at 70% introduces noise from stylistic or irrelevant tokens. This component enables scalable, interpretable, and task-general supervision from natural language feedback.

### 3.4 REWARD MODEL LEARNING

To enable NL-Gradient optimization, we train a reward model $R_\phi$ that jointly generates natural language critiques and structured span-level feedback in a single autoregressive pass. Instead of predicting scalar scores, we frame reward modeling as a text generation task—producing both natural language evaluations and span-level labels as output sequences.

**Model Objective.**  Given a prompt $x$ and model response $y = (y_1, \ldots, y_T)$, the reward model outputs a sequence $z = [c; \mathcal{A}(y)]$, where $c$ is a critique and $\mathcal{A}(y)$ is a JSON-formatted map labeling spans in $y$ as $\texttt{positive}$, or $\texttt{negative}$. We model this as conditional language generation: $p_\phi(z \mid x, y) = \prod_{t=1}^{|z|} p_\phi(z_t \mid z_{<t}, x, y)$, and optimize via maximum likelihood with a cross-entropy loss: $\mathcal{L}_R(\phi) = -\mathbb{E}_{(x,y,z) \in \mathcal{D}_R} \left[ \log p_\phi(z \mid x, y) \right]$.

This formulation provides three advantages: (1) flexibility across tasks via textual supervision; (2) fine-grained gradient flow through tokenized outputs; and (3) interpretable feedback combining explanation and token-level reward in one model. Each training instance is serialized as $[x; y; z]$, and the model is fine-tuned using teacher forcing under a standard causal LM objective. This unified, text-based approach simplifies the pipeline while enabling both structured and natural language feedback to drive token-level learning in TEXT2GRAD.

### 3.5 NL-GRADIENT POLICY OPTIMIZATION

Traditional RL methods rely on sequence-level scalar rewards, which obscure token-level credit assignment and limit precision. This is especially problematic in tasks like summarization and code generation, where only specific parts of the output may be incorrect. To address this, TEXT2GRAD uses dense token-level pseudo-rewards $\{\delta_t\}$ derived from structured textual feedback to enable fine-grained advantage estimation: $A_t = \sum_{l=0}^{T-t-1} (\gamma\lambda)^l \delta_{t+l}^{\text{TD}}$,  where $\delta_t^{\text{TD}} = r_t^{\text{total}, A} + \gamma V_\psi(x, y_{<t+1}) - V_\psi(x, y_{<t})$, where $\gamma$ is the discount factor, $\lambda$ is the GAE parameter, $V_\psi$ is the value function, and $r_t^{\text{total}, A} = \delta_t + r_t^{\text{KL}}$ combines the token-level pseudo-reward $\delta_t$ with the KL penalty term $r_t^{\text{KL}}$.

Given a response $y$, we query the trained reward model $R_\phi$ to generate a natural language critique and span-level reward map, which is parsed into token-wise rewards $\{\delta_t\}_{t=1}^T$. These are used to construct

the *NL-Gradient*: $g_{\text{NL}} = \sum_{t=1}^{T} \delta_t \cdot \nabla_\theta \log \pi_\theta(y_t \mid x, y_{<t})$, where $\pi_\theta$ is the policy parameterized by $\theta$, providing localized learning signals aligned with feedback.

We then compute token-level advantages using GAE and integrate them into the PPO objective:

$$L^{\text{PPO}}(\theta) = \mathbb{E}_t\left[\min\left(\rho_t A_t, \ \text{clip}(\rho_t, 1 - \epsilon, 1 + \epsilon)A_t\right)\right] - \beta \, \mathcal{H}\left(\pi_\theta(\cdot \mid x, y_{<t})\right),$$

where $\rho_t = \pi_\theta(y_t \mid x, y_{<t})/\pi_{\theta_{\text{old}}}(y_t \mid x, y_{<t})$ is the importance ratio, $\mathcal{H}$ is the entropy bonus, $\beta$ is the entropy coefficient, and $\epsilon$ is the clipping threshold that stabilizes updates by constraining large policy shifts. By transforming natural language feedback into token-level gradients, TEXT2GRAD enables interpretable, precise, and efficient policy optimization.

### 3.6 THEORETICAL ANALYSIS: DISCRIMINATIVE POWER OF TOKEN-LEVEL REWARDS

Our analysis shows that token-level rewards derived from textual feedback lead to sharper and more discriminative advantage estimates than end-of-sequence rewards. Under our formulation, the advantage at timestep $t$ is computed as $A_t^A = \sum_{k=t}^{T} (\gamma\lambda)^{k-t} \delta_k$, where $\delta_k$ are pseudo-rewards aligned to tokens via natural language critiques. In contrast, end-of-sequence rewards yield $A_t^B = (\gamma\lambda)^{T-t} \sum_{k=t}^{T} \delta_k$, discounting all feedback uniformly. The difference in temporal credit assignment is given by $\Delta A_t^A - \Delta A_t^B = \sum_{k=t}^{T-1} (\gamma\lambda)^{k-t} \Delta\delta_k$, which amplifies early feedback differences. For typical settings where $\gamma\lambda \approx 0.95$, a token-level reward at step $k = T - 20$ is weighted nearly $0.95^{-20} \approx 2.8$ times more than it would be under end-of-sequence supervision—showing that natural language-guided token-level feedback is nearly 3× more effective for early credit assignment. This yields more informative gradients and improves the policy's ability to localize and correct errors in long-form outputs. The full derivation and comparison are provided in Appendix A.

## 4 EXPERIMENTS

We evaluate TEXT2GRAD on summarization, code generation, and question answering to test its ability to transform natural language feedback into fine-grained policy updates. Our experiments demonstrate that TEXT2GRAD outperforms scalar-reward baselines such as PPO, with improved sample efficiency, faster convergence, and better accuracy.

### 4.1 DATASETS OVERVIEW

**SLF5K** (Scheurer et al., 2023): A summarization dataset with 5,000 Reddit posts, human-written summaries, and feedback. We use all 5,000 samples for SFT, reward modeling, and policy training, with 500 for evaluation. **KodCode** (Xu et al., 2025): A code generation benchmark with 447K question–solution–test triplets across 12 domains. We sample 9K GPT-4o completions to train the reward model, and use KodCode-Light-RL-10k for policy optimization. **UltraFeedback** (Cui et al., 2023): A QA dataset with 64K prompts and 256K completions from 17 models. Following Huang et al. (2024), we split the data into 30% SFT, 50% reward modeling, and 20% RL.

### 4.2 REWARD MODEL EVALUATION

A core component of TEXT2GRAD is the unified reward model, trained to emulate the evaluative reasoning of advanced LLMs (i.e., GPT-4o) by producing structured, token-level feedback.

**Experimental Setup** We fine-tune `Llama3.1-8B-Instruct` (Grattafiori et al., 2024) to serve as the reward model across all tasks. It is trained to output both a natural language critique and a span-level reward signal, using supervision generated by GPT-4o. To ensure high-quality labels, we use a CoT prompting strategy (Wei et al., 2022; Ding et al., 2024) in which GPT-4o first reasons through the correctness of a model response, then articulates strengths and weaknesses, and finally highlights token spans as `positive` or `negative`. This structured annotation improves feedback precision and interpretability, enabling richer training signals than scalar-only supervision.

**Main Results** Table 1 presents token-level precision and recall for feedback identification, along with span-level win rates in pairwise comparisons (with vs. without CoT reasoning) and human-alignment accuracy. To compute token-level metrics, we map each annotated span to its constituent tokens: tokens within `positive` spans are labeled $+1$, `negative` spans $-1$, and all others $0$

Table 1: Quantitative evaluation of reward models with and without CoT prompting, measured by span-level precision/recall, preference win-rate (W:T:L), and human annotation accuracy.

| Dataset | Positive Token | | Negative Token | | Win-Rate (W:T:L) | Human-anno Acc. |
|---|---|---|---|---|---|---|
| | Prec. | Rec. | Prec. | Rec. | | |
| SLF5K | 0.58 | 0.63 | 0.58 | 0.43 | 62:9:29 | 86% |
| *No CoT* | 0.63 | 0.46 | 0.53 | 0.40 | – | – |
| UltraFeedback | 0.66 | 0.43 | 0.46 | 0.22 | 53:9:38 | 82% |
| *No CoT* | 0.61 | 0.59 | 0.40 | 0.35 | – | – |
| KodCode | 0.64 | 0.68 | 0.84 | 0.71 | 72:7:21 | 94% |
| *No CoT* | 0.62 | 0.61 | 0.75 | 0.78 | – | – |

(neutral); model predictions are evaluated against this derived ground truth. Span-level recall is measured through Exact/Partial Match metrics (Appendix C). Our annotation pipeline ensures high fidelity with unmatched-span rates below 2.5% across all datasets (Table 15), confirming that reward signals are grounded in actual model outputs.

Across all datasets, the CoT-based reward model consistently outperforms the ablated variant, achieving a 62% win rate on SLF5K and **86%** alignment with human annotations. Although the precision for positive spans slightly decreases (58% vs. 63%), the recall improves significantly (63% vs. 46%), indicating better coverage and reduced overfitting to surface-level cues. The moderate negative-token recall (22% on UltraFeedback, 43% on SLF5K) reflects label imbalance: approximately 63–70% of tokens are neutral, so only a minority receive non-zero rewards. From a policy-learning perspective, precision is more critical than recall, as false-signed rewards directly corrupt gradients, while missing correct tokens merely reduces update density. Our precision-first design, combined with high human alignment (>82%), produces stable advantages and substantial policy improvements despite moderate recall. Similar trends hold on UltraFeedback and KodCode, with robust performance in the code domain (KodCode win rate: 72%). Critically, our pipeline enforces strict critique–span alignment: every labeled span must be justified by prior CoT reasoning (Appendix B), and post-processing ensures spans are exact response quotes (unmatched rate <2.5%, Table 15). This produces high consistency (82–94% human alignment), enabling scalable and high-fidelity feedback without signal loss. Appendix K further shows that the training time overhead is modest compared to PPO, primarily due to a single reward model forward pass per trajectory. Detailed human evaluation results are provided in Appendix M.

Collectively, these results demonstrate that **structured natural language reasoning, coupled with precise span-level grounding, enables accurate, discriminative, and data-efficient reward modeling** forming a robust foundation for token-level policy learning in TEXT2GRAD. The pairwise-comparison prompt is provided in Appendix E. Additional metrics are reported in Appendix C.

### 4.3 SLF5K (SCHEURER ET AL., 2023): SUMMARIZATION

We evaluate TEXT2GRAD on the SLF5K dataset (Scheurer et al., 2023), which involves generating summaries of Reddit posts that closely align with human-written references. This task provides natural language feedback and span-level annotations, making it well-suited for evaluating the effectiveness of token-level reward modeling. Additional hyperparameters are provided in Appendix D.

**Experimental Setup** We use `Llama3.1-8B-Instruct` (Grattafiori et al., 2024) as the base policy model. It is first fine-tuned using supervised learning on SLF5K to control output length and content coverage, and subsequently optimized using our NL-Gradient method. We compare TEXT2GRAD against several baselines: (1) PPO (Schulman et al., 2017) trained with scalar rewards, (2) DPO (Rafailov et al., 2023) for preference optimization, (3) PRM-PPO (Lightman et al., 2023) combining preference modeling with PPO, (4) supervised fine-tuning (SFT), and (5) SFT enhanced with reward-guided reflection (Shinn et al., 2023; Madaan et al., 2023). Appendix N details how PRM spans were defined for each domain, and includes the GPT-3.5 and GPT-4o outputs as reference points. Evaluation metrics include ROUGE (Lin, 2004), BLEU (Papineni et al., 2002), BERTScore (Zhang et al., 2019), and LLM-as-a-Judge (Gu et al., 2024).
**Main Results** As shown in Table 2, TEXT2GRAD achieves SOTA performance on all metrics, outperforming scalar-reward and reflection-based baselines. It surpasses PPO by **+25.3% BLEU**

Table 2: Performance comparison on the SLF5K summarization dataset. The policy model is Llama-3.1-8B-Instruct. **Bold** indicates best results; underline indicates second best.

| Method | R-1 | R-2 | R-L | BLEU | BERTScore |
|---|---|---|---|---|---|
| *Proprietary Models* | | | | | |
| ChatGPT-3.5 | 0.155 | 0.059 | 0.108 | 0.020 | 0.844 |
| GPT-4o | 0.296 | 0.066 | 0.203 | 0.030 | 0.886 |
| *Open-Source Baselines (8B)* | | | | | |
| SFT | 0.285 | 0.078 | 0.195 | 0.032 | 0.875 |
| SFT + Reflection | 0.329 | 0.087 | 0.225 | 0.041 | 0.888 |
| DPO | 0.327 | 0.101 | 0.224 | 0.039 | 0.885 |
| PPO | 0.365 | 0.132 | 0.262 | 0.075 | 0.893 |
| PRM-PPO | 0.341 | 0.130 | 0.254 | 0.069 | 0.889 |
| ILF (Scheurer et al., 2023) | 0.349 | 0.134 | 0.259 | 0.073 | 0.892 |
| *Ours* | | | | | |
| TEXT2GRAD (w/o CoT) | 0.380 | 0.140 | 0.275 | 0.085 | 0.898 |
| TEXT2GRAD | **0.400** | **0.155** | **0.291** | **0.094** | **0.902** |

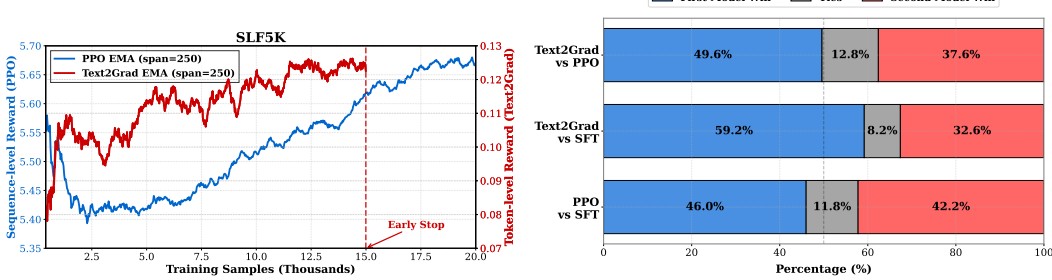

(a) Reward curve for SLF5K dataset.  (b) GPT4 Judge Comparison across Methods.

Figure 3: Combined figure for SLF5K dataset analysis.

and **+6.7 ROUGE-L**, exceeds DPO and PRM-PPO by +6.7 and +3.7 ROUGE-L respectively, and improves over SFT+Reflection by **+3.3 ROUGE-L**, confirming that *gradient-based internalization* of feedback yields stronger gains than *inference-time correction*. To validate our span-based design, we compare against dense token-level labeling. Despite maximal supervision, dense labeling performs substantially worse (ROUGE-L: 0.196 vs. 0.291), as it labels ~70% of tokens predominantly on function words rather than semantic spans, introducing noise into advantage estimates. Our span-based approach achieves superior performance while maintaining high grounding fidelity (unmatched rate <2.5%) and reducing annotation costs by 85–90% (Appendix I.1). Qualitatively, GPT-4-as-a-Judge preferences (Figure 3b) show a **12% win-rate gain over PPO**, indicating more coherent and informative outputs. Quantitatively, Figure 3a reveals TEXT2GRAD converges **22% faster**, demonstrating that token-level gradients accelerate learning while allowing interpretable updates. The table also shows that removing reasoning degrades performance.

## 4.4 KODCODE (XU ET AL., 2025): CODE GENERATION

We evaluate TEXT2GRAD on the KodCode dataset (Xu et al., 2025), which focuses on generating correct Python solutions across 12 diverse problem domains. This task highlights the importance of span-level feedback in structured text generation, where subtle errors can invalidate the entire output.

**Experimental Setup**  We adopt `Llama3.1-8B-Instruct` (Grattafiori et al., 2024) as the policy model. For reward model training, we sample 10,000 prompt–completion pairs from the supervised dataset, using GPT-4o outputs as high-quality references and GPT-3.5 completions as challenging negatives to form pairwise comparisons. Annotations include textual critiques and span-level labels. We benchmark TEXT2GRAD against PPO (Schulman et al., 2017) and strong baselines, evaluating

via pass@1 accuracy on HumanEval (Chen et al., 2021), MBPP (Austin et al., 2021), and their robustness-enhanced variants, HumanEval+ and MBPP+ (Yu et al., 2024).

Table 3: Code generation benchmarks (pass@1 %). **Bold**: best; underline: second best.

| Method | Size | HumanEval | | MBPP | | Avg. |
| --- | --- | --- | --- | --- | --- | --- |
| | | Base | Plus | Base | Plus | |
| *Proprietary & Pre-trained Models* | | | | | | |
| Llama-3.2-Instruct | 3B | 48.2 | 43.9 | 61.9 | 51.3 | 51.3 |
| Llama-3.1-Instruct | 8B | 64.0 | 58.5 | 66.7 | 55.0 | 61.1 |
| CodeLlama | 34B | 51.8 | 43.9 | 69.3 | 56.3 | 55.3 |
| Gemini Pro | – | 63.4 | 55.5 | 72.9 | 57.9 | 62.4 |
| *Fine-tuned (Llama-3.1-8B-Instruct)* | | | | | | |
| DPO | 8B | 65.2 | 56.7 | 66.1 | 56.1 | 61.0 |
| PPO | 8B | 64.6 | 61.0 | 68.5 | 55.8 | 62.5 |
| PRM-PPO | 8B | 61.5 | 59.8 | 65.1 | 54.9 | 60.3 |
| ILF | 8B | 63.4 | 60.4 | 68.5 | 57.1 | 62.3 |
| *Ours* | | | | | | |
| TEXT2GRAD (w/o CoT) | 8B | 63.8 | 57.3 | 62.2 | 53.4 | 59.2 |
| TEXT2GRAD | 8B | **67.7** | **61.6** | **73.3** | **61.6** | **66.1** |

**Main Results** Table 3 shows TEXT2GRAD outperforms all baselines—both pre-trained and fine-tuned—across all benchmarks. Against PPO, it gains **+5.8** on MBPP+ and **+3.6** on HumanEval+, demonstrating superior robustness to adversarial test cases. It also surpasses DPO and PRM-PPO by **5.1** and **5.8** average points, respectively. Critically, the ablated variant (TEXT2GRAD w/o CoT) underperforms by **6.9** points on average, confirming that structured natural language feedback—not just span labels—is essential for effective token-level credit assignment. These results validate that TEXT2GRAD precisely localizes and corrects subtle coding errors, yielding programs that generalize reliably under stress.

### 4.5 ULTRAFEEDBACK (CUI ET AL., 2023): OPEN-DOMAIN QUESTION ANSWERING

To evaluate TEXT2GRAD on general-purpose alignment, we test it on UltraFeedback (Cui et al., 2023), a diverse QA benchmark spanning multiple domains and difficulty levels. This task assesses generalization to open-ended prompts, factual accuracy, and conversational coherence.

**Experimental Setup** We use `Llama3-8B-Instruct` as the policy backbone. Evaluation metrics include: (1) **AlpacaEval 2.0** (Dubois et al., 2024) for instruction alignment, (2) **ARC-Challenge** (Clark et al., 2018) for reasoning, and (3) **MT-Bench** (Zheng et al., 2023) for multi-turn dialogue quality. UltraFeedback responses are long and do not contain explicit intermediate reasoning steps, making PRM-style step annotation ill-defined and costly (6–8× higher token budget). We therefore omit PRM-PPO; see Appendix N for details.

Table 4: UltraFeedback QA benchmarks. **Bold**: best; underline: second best.

| Method | AlpacaEval | ARC-C | MT-Bench |
| --- | --- | --- | --- |
| *Proprietary Models* | | | |
| GPT-4 | 30.2 | **96.4** | 7.93 |
| GPT-3.5 | 22.7 | 85.2 | 6.91 |
| *Open-Source (Llama3-8B-Instruct)* | | | |
| Base | 22.6 | 80.5 | 6.87 |
| DPO | 32.6 | 81.0 | 7.01 |
| PPO | 32.4 | 82.7 | 7.43 |
| ILF | 30.1 | 80.9 | 7.08 |
| *Ours* | | | |
| TEXT2GRAD (w/o CoT) | 28.6 | 83.1 | 7.49 |
| TEXT2GRAD | **34.7** | 84.4 | **7.58** |

**Main Results** As shown in Table 4, TEXT2GRAD consistently improves over both the base model and PPO across all metrics. On AlpacaEval 2.0, TEXT2GRAD achieves a 12.1-point gain over the base model and a 2.3-point improvement over PPO, indicating stronger instruction alignment and preference satisfaction. On ARC-Challenge, TEXT2GRAD shows improved reasoning over base and PPO, while MT-Bench results show better multi-turn dialogue performance.

Our ablation study isolates the effect of CoT reasoning in the annotation pipeline. Specifically, we remove CoT and provide feedback only as span-level scores without natural-language explanations. Training without CoT yields consistent drops across all metrics, with the largest decrease on AlpacaEval. This suggests that natural-language explanations produce more actionable token-level supervision for NL-Gradient optimization.

## 4.6 CASE STUDY

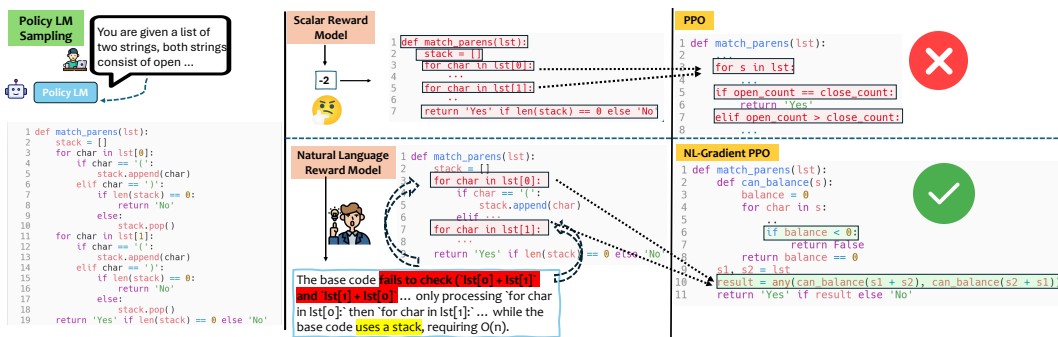

Figure 4: A case study from the code generation scenario comparing PPO vs. TEXT2GRAD.

Figure 4 shows how TEXT2GRAD corrects a faulty implementation of `match_parens` while standard PPO fails. The policy LM first produces a buggy patch. A scalar reward model gives PPO a single negative score (–2), leaving the optimizer without guidance on where the error is located. After several updates, it still ignores the two cross–concatenation checks required by hidden tests.

TEXT2GRAD proceeds differently. The natural language reward model highlights the exact faulty span `for char in lst[0] ...` and explains that the code "*fails to check `lst[0] + lst[1]` and `lst[1] + lst[0]`.*" This critique is aligned with the offending tokens and converted into negative rewards for that span and positive rewards for the rest. A single NL–Gradient update rewrites only the highlighted lines. The resulting function passes all unit tests. This example underscores the advantages of TEXT2GRAD. Additional qualitative results appear in Appendix F.

## 4.7 CROSS-MODEL GENERALIZATION

To validate that TEXT2GRAD's gains transfer across model families, we evaluated on Mistral-7B-Instruct-v0.2 across code generation, open-domain QA, and summarization. As shown in Table 5, TEXT2GRAD consistently outperforms baselines across all tasks: on code generation, average pass@1 improves from 42.9 (DPO) to 45.3; on QA, AlpacaEval increases from 26.17 to 29.40; on summarization, ROUGE-L reaches 0.24. These results confirm the method's generality.

Table 5: Cross-model evaluation on Mistral-7B-Instruct-v0.2. **Bold**: best per metric.

| Method | Code (pass@1 %) | | | | QA | | Summarization | | |
|---|---|---|---|---|---|---|---|---|---|
| | HE | HE+ | MBPP | MBPP+ | AlpacaEval | MT-Bench | R-L | BLEU | BERT |
| Base | 42.1 | 36.0 | 44.7 | 37.0 | 17.11 | 6.60 | 0.200 | 0.024 | 0.66 |
| PPO | 45.7 | 38.4 | 47.1 | 38.3 | 19.62 | 6.55 | 0.230 | 0.035 | 0.70 |
| DPO | 47.6 | 39.6 | 46.7 | 37.6 | 26.17 | 6.30 | 0.210 | 0.030 | 0.69 |
| TEXT2GRAD | **50.0** | **40.9** | **49.6** | **40.6** | **29.40** | **6.78** | **0.240** | **0.041** | **0.72** |

## 5 CONCLUSION

We presented TEXT2GRAD, a new framework for learning from natural language feedback by converting free-form textual critiques into span-level reward signals and actionable gradients. Unlike traditional RLHF approaches that rely on scalar rewards or inference-time prompting strategies, TEXT2GRAD directly incorporates feedback into the training process through token-aware policy updates. This enables precise credit assignment and more interpretable learning dynamics. Experimental results across summarization, code generation, and question answering demonstrate that TEXT2GRAD consistently outperforms scalar-reward PPO and prompt-based baselines in both alignment quality and sample efficiency. Cross-model evaluation on Mistral-7B-Instruct-v0.2 (Section 4.7) further validates that these gains transfer across model families and architectures. Overall, TEXT2GRAD opens a new direction for fine-grained, feedback-driven optimization of LLMs, moving beyond scalar supervision toward more human-like, interpretable, and effective learning.

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

## A  DISCRIMINATIVE POWER OF TOKEN-LEVEL REWARDS

A key design choice in our method is to provide dense, token-level feedback rather than sparse, end-of-sequence rewards. Intuitively, localized reward signals allow the policy to attribute credit or blame more precisely to specific parts of the output. In this section, we formalize this intuition and show how token-level rewards lead to sharper and more discriminative advantage estimates, thereby improving policy learning.

**Background.**  In reinforcement learning, policy updates are guided by the advantage function, which measures how much better (or worse) an action is compared to the policy's expected value. Using Generalized Advantage Estimation (GAE), the advantage at timestep $t$ is computed from the temporal-difference (TD) errors:

$$A_t = \sum_{l=0}^{T-t} (\gamma\lambda)^l \delta_{t+l}, \quad \text{where} \quad \delta_t = r_t + \gamma V(s_{t+1}) - V(s_t),$$

and $V$ is the value function, $\gamma$ is the discount factor, and $\lambda$ is the GAE parameter.

**Comparing Token-Level vs. End-of-Sequence Reward Settings.**  We define two settings for reward assignment:

**Setting A: Token-Level Rewards.** Each token may receive its own feedback:

- $r_t^{\text{token,A}} \neq 0$ for many $t \in [1, T]$
- Total reward: $r_t^{\text{total,A}} = r_t^{\text{token,A}} + r_t^{\text{KL}}$

**Setting B: End-of-Sequence Reward.** Only the final token is rewarded:

- $r_t^{\text{token,B}} = 0$ for all $t < T$
- $r_T^{\text{token,B}} \neq 0$; total reward: $r_t^{\text{total,B}} = r_t^{\text{token,B}} + r_t^{\text{KL}}$

Let $\tau_1$ and $\tau_2$ be two trajectories, where $\tau_1$ is qualitatively better than $\tau_2$. Define $\Delta r_t = r_t^{\text{token,A}}(\tau_1) - r_t^{\text{token,A}}(\tau_2)$, and assume all KL terms and value functions are held constant for simplicity (the general case follows similarly).

**Advantage Difference Across Trajectories.**  The advantage difference under each setting is:

$$\Delta A_t^A = \sum_{k=t}^{T} (\gamma\lambda)^{k-t} \Delta r_k, \quad \Delta A_t^B = (\gamma\lambda)^{T-t} \sum_{k=t}^{T} \Delta r_k.$$

Even if $\sum_{k=t}^{T} \Delta r_k$ is the same in both cases (i.e., the same total reward difference), $\Delta A_t^A > \Delta A_t^B$ whenever any $\Delta r_k > 0$ for $k < T$, because:

$$(\gamma\lambda)^{k-t} > (\gamma\lambda)^{T-t}, \quad \text{for all } k < T.$$

This means the earlier the reward difference occurs in the sequence, the more strongly it is emphasized in Setting A relative to Setting B.

**Amplification of Early Signal.**  To quantify this difference, define the amplification factor:

$$\alpha(k, T) = \frac{(\gamma\lambda)^{k-t}}{(\gamma\lambda)^{T-t}} = (\gamma\lambda)^{-(T-k)}.$$

For a typical value of $\gamma\lambda = 0.95$ and a gap of $T - k = 20$ steps (i.e., the difference occurs 20 tokens before the final token), we have:

$$\alpha(k, T) \approx 0.95^{-20} \approx 2.8,$$

meaning that in Setting A, the advantage function weights early reward differences nearly 3× more than in Setting B.

This analysis confirms that token-level feedback improves the discriminative power of the advantage signal: even if the total reward difference is the same, Setting A assigns more importance to earlier deviations in quality. This sharper signal allows the policy to learn localized corrections—e.g., improving grammar or factual consistency in specific parts of a summary—rather than attributing success or failure to the entire sequence. As a result, our method enables faster convergence and better fine-tuning, especially on open-ended tasks where quality varies across tokens.

## B  GPT-4O CHAIN-OF-THOUGHT ANNOTATION PROMPTS

This section presents the detailed prompt templates used for generating dual-feedback annotations across our three experimental datasets. Each prompt is designed to elicit both natural language critiques and structured span-level feedback through CoT reasoning.

### B.1  SLF5K DATASET

The following prompt template is used for generating annotations on the SLF5K summarization dataset:

---

Listing 1: SLF5K GPT-4o Annotation Prompt

```
1  Please critique the following summary of a post and provide
       feedback in the specified JSON format:
2
3  ---
4
5  **Original Post:**
6  {post}
7
8  **Generated Summary:**
9  {generated_summary}
10
11 ---
12
13 **Definitions:**
14 - **good_spans**: 0-2 phrases from the summary that greatly
       improve its quality by accurately and concisely
       capturing the original post's core meaning or key
       details, as explained in 'textual_feedback'. Empty if
       none apply.
15 - **poor_spans**: 0-2 phrases from the summary that
       noticeably harm its quality due to inaccuracy,
       redundancy, poor wording, or being less important and
       replaceable with more critical content, as explained in
       'textual_feedback'. Empty if none apply.
16
17 ---
18
19 **Instructions:**
20 1. Identify the summary's most essential strengths that
       reflect the original post accurately and its most
       critical weaknesses that misrepresent or confuse it.
```

```
21  2. Select 0-2 of the most significant phrases for
        'good_spans' and 'poor_spans', keeping them concise and
        impactful, with brief justifications. Include none if no
        phrases stand out.
22  3. Ensure 'good_spans' and 'poor_spans' are directly
        supported by the analysis in 'textual_feedback'.
23
24  ---
25
26  **Chain of Thought:**
27  First, carefully analyze both the original post and the
        generated summary:
28  1. What are the key points of the original post?
29  2. Which of these key points are accurately captured in the
        summary?
30  3. What important information is missing from the summary?
31  4. Are there any inaccuracies or misrepresentations in the
        summary?
32  5. Which specific phrases in the summary represent its
        strongest elements?
33  6. Which specific phrases in the summary represent its
        weakest elements?
34
35  Based on this analysis, formulate your textual feedback and
        identify the good and poor spans.
36
37  ---
38
39  **Output Format:**
40  Provide a concise, one-paragraph critique and the GOOD/POOR
        spans in this JSON structure:
41  ```json
42  {
43    "textual_feedback": "Your critique here summarizing key
          strengths and weaknesses in one paragraph.",
44    "good_spans": ["phrase1", "phrase2"],  // 0-2 concise
          phrases from the generated summary, tied to
          textual_feedback, or [] if none
45    "poor_spans": ["phrase1", "phrase2"]   // 0-2 concise
          phrases from the generated summary, tied to
          textual_feedback, or [] if none
46  }
47  ```
48
49  Focus on precision: include only the most impactful phrases
        of the generated summary, avoiding excessive or minor
        details.
```

## B.2 ULTRAFEEDBACK DATASET

The following prompt template is used for generating annotations on the UltraFeedback question-answering dataset:

Listing 2: UltraFeedback GPT-4o Annotation Prompt

```
1  <CritiquePrompt>
2
3      <Instructions>Critique a response to a user input and
           provide feedback in JSON format:</Instructions>
4
5
6      <EvaluationCriteria>
7          <Criterion name="Accuracy">Does it correctly address
               the input?</Criterion>
8          <Criterion name="Relevance">Does it stay on
               topic?</Criterion>
9          <Criterion name="Clarity">Is it easy to
               understand?</Criterion>
10          <Criterion name="Completeness">Does it cover the
               input's core needs?</Criterion>
11      </EvaluationCriteria>
12
13
14      <SpanGuidelines>
15          <GoodSpans>
16              <Description>Phrases from the response that best
                   capture its strengths (e.g., accurate,
                   relevant, clear). Select only the most
                   essential and impactful phrases, directly
                   tied to 'textual_feedback'.</Description>
17          </GoodSpans>
18          <PoorSpans>
19              <Description>Phrases from the response that best
                   highlight its weaknesses (e.g., inaccurate,
                   irrelevant, vague). Select only the most
                   essential and impactful phrases, directly
                   tied to 'textual_feedback'.</Description>
20          </PoorSpans>
21          <Requirement>Spans must be exact quotes from the
               response.</Requirement>
22      </SpanGuidelines>
23
24
25      <ReflectionProcess>
26          <Step>First, carefully analyze the user input to
               understand the core question or request.</Step>
27          <Step>Next, examine the generated response against
               each evaluation criterion.</Step>
28          <Step>For each criterion, identify specific
               strengths and weaknesses with supporting
               evidence from the response.</Step>
29          <Step>Consider how well the response addresses the
               user's explicit and implicit needs.</Step>
30          <Step>Finally, synthesize your analysis into a
               coherent critique that highlights the most
               important points.</Step>
31      </ReflectionProcess>
32
33
34      <Separator>---</Separator>
35
36
```

```
37      <UserInput>{entry['prompt']}</UserInput>
38      <GeneratedResponse>{entry['response']}</GeneratedResponse>
39
40
41      <Separator>---</Separator>
42
43
44      <OutputFormat>
45          <Description>Provide the critique in the following
                JSON structure:</Description>
46          <JSONExample>
47              {{
48                  "textual_feedback": "One-paragraph critique
                        summarizing strengths and weaknesses,
                        tied to spans.",
49                  "good_spans": ["phrase1", "phrase2", ...],
                        // Impactful phrases from
                        <GeneratedResponse>, or [] if none
50                  "poor_spans": ["phrase1", "phrase2", ...]
                        // Impactful phrases from
                        <GeneratedResponse>, or [] if none
51              }}
52          </JSONExample>
53      </OutputFormat>
54
55  </CritiquePrompt>
```

## B.3 KODCODE DATASET

The following prompt template is used for generating annotations on the KodCode code generation dataset:

Listing 3: KodCode GPT-4o Annotation Prompt

```
1  Analyze the following code solution for the given problem:
2
3  Problem Description:
4  '''
5  {problem}
6  '''
7
8  Submitted Code:
9  '''
10 {solution}
11 '''
12
13 Test Results:
14 Passed: {passed}
15
16 {%- if not passed -%}
17 Test Question:
18 {test_question}
19
20 Error Output:
21 {stdout}
```

```
22  {%- endif -%}
23
24  Please analyze the code and identify the following in JSON
       format:
25
26  1. Identify any error-causing code segments directly from
       the submitted solution.
27  2. Provide detailed feedback on the code's functionality,
       issues, and improvement suggestions.
28     - First, understand what the code is trying to accomplish
29     - Analyze the algorithm and approach used
30     - Identify any logical errors or inefficiencies
31     - Consider edge cases and potential improvements
32  3. Point out any code segments from the solution that work
       but could be improved.
33
34  Return your analysis in this JSON structure:
35  ```json
36  {
37      "Code Feedback": "Provide a detailed explanation of the
           code's functionality, any potential issues, and
           suggestions for improvement. Use markdown formatting
           for better readability.",
38      "wrong_code": ["Extract ONLY the problematic code
           segments FROM THE SUBMITTED SOLUTION that cause
           failures. Must be exact quotes. Leave empty [] if
           none found."],
39      "improvement_code": ["Extract ONLY the working but
           improvable code segments FROM THE SUBMITTED
           SOLUTION. Must be exact quotes. Leave empty [] if
           none needed."]
40  }
41  ```
42  Note: For 'wrong_code' and 'improvement_code', only include
       direct quotes from the submitted code above, not
       suggested fixes."""
```

## C  ADDITIONAL REWARD MODEL PERFORMANCE RESULTS

This section presents supplementary evaluation metrics for the reward model across three datasets, focusing on span prediction quality (UltraFeedback), code suggestion fidelity (KodCode), and language modeling coherence (SLF5K). These metrics quantify whether feedback actually conditions behavior at the criticized spans, demonstrating that span-level rewards reliably localize and influence the regions referenced in critiques.

Table 6: Span prediction performance on the UltraFeedback dataset. Reported as **GT / Pred (Exact - Partial)**, where GT = ground truth span count, Pred = predicted span count. Exact and Partial denote exact and partially overlapping matches, respectively. OUI (Overlap Unit Index) quantifies boundary alignment precision.

| Span Type | GOOD Spans | POOR Spans |
|---|---|---|
| Count (GT / Pred) | 1150 / 932 | 812 / 528 |
| Match (Exact - Partial) | (394 - 468) | (167 - 274) |
| OUI | 0.40 | 0.27 |

Table 7: Code suggestion quality on the KodCode dataset. Exact Match measures the proportion of generated suggestions that exactly match the reference code. $\geq 90\%$ Overlap evaluates the proportion of suggestions with at least 90% overlap with ground-truth code segments.

| Suggestion Type | Exact Match | $\geq 90\%$ Overlap |
|---|---|---|
| Improvement Suggestions | 0.47 | 0.67 |
| Error Correction | 0.55 | 0.79 |

Table 8: Extended language modeling metrics on the SLF5K dataset. Human reference perplexity: 37.375. **Bold**: best results.

| Method | Perplexity↓ | BERTScore↑ | |
|---|---|---|---|
| | | Precision | Recall |
| *Proprietary Models* | | | |
| ChatGPT-3.5 | 27.288 | 0.806 | 0.884 |
| GPT-4o | 53.242 | 0.879 | 0.894 |
| *Open-Source (Llama-3.1-8B-Instruct)* | | | |
| Base | 19.248 | 0.848 | 0.894 |
| SFT | 44.103 | 0.865 | 0.885 |
| SFT + Reflection | 34.823 | 0.880 | 0.897 |
| PPO | 28.472 | 0.892 | 0.895 |
| *Ours* | | | |
| TEXT2GRAD | **25.423** | **0.903** | **0.902** |

## D    TRAINING HYPERPARAMETERS

This section provides complete hyperparameters for both reward model training and policy optimization.

### D.1    REWARD MODEL TRAINING

We fine-tune Llama-3.1-8B-Instruct as the reward model using LoRA for parameter-efficient adaptation.

Table 9: Hyperparameters for reward model training (shared across all datasets).

| Parameter | Value | Parameter | Value |
|---|---|---|---|
| Base Model | Llama-3.1-8B | Learning Rate | $1 \times 10^{-5}$ |
| Hardware | 8×V100 32G | Training Epochs | 2 |
| Parallelism | ZeRO-3 | Optimizer | AdamW |
| Global Batch Size | 8 | Precision | FP16 |
| LoRA Rank ($r$) | 16 | LoRA Dropout | 0.1 |
| LoRA Alpha ($\alpha$) | 32 | Grad. Clip Norm | 1.0 |
| LoRA Targets | `q_proj, v_proj` | Weight Decay | $3 \times 10^{-7}$ |

### D.2    NL-GRADIENT PPO OPTIMIZATION

Hyperparameters are tailored per dataset to balance training stability and optimization efficiency.

Table 10: Hyperparameters for NL-Gradient PPO across all datasets.

| Hyperparameter | SLF5K | UltraFeedback | KodCode |
|---|---|---|---|
| *Model Configuration* | | | |
| Policy Model | Llama 3.1 8B | Llama 3 8B | Llama 3.1 8B |
| Reward Model | Llama 3.1 8B | Llama 3.1 8B | Llama 3.1 8B |
| *Optimization* | | | |
| Learning Rate | $1 \times 10^{-6}$ | $1 \times 10^{-6}$ | $5 \times 10^{-7}$ |
| LR Scheduler | Linear | Cosine | Cosine |
| Training Epochs | 4 | 4 | 4 |
| *Batch Settings* | | | |
| Global Batch Size | 12 | 8 | 8 |
| Mini-batch Size | 1 | 1 | 1 |
| PPO Epochs/Batch | 4 | 4 | 4 |
| Gradient Accum. | 12 | 8 | 8 |
| *KL Regularization* | | | |
| Initial KL Coef. | 0.2 | 0.05 | 0.07 |
| Target KL | 6.0 | 3.0 | 1.0 |
| KL Penalty Type | Full | Full | Full |
| Adaptive KL | ✓ | ✓ | ✓ |
| *Infrastructure* | | | |
| Hardware | 8×V100 80G | 8×A100 80G | 8×A100 80G |
| Parallelism | ZeRO-3 | ZeRO-3 | ZeRO-1 |

**Notes:** SLF5K uses linear scheduler; UltraFeedback adopts conservative KL for diverse responses; KodCode uses strictest KL target (1.0) to preserve code semantics.

## E  GPT-4O JUDGE COT INFLUENCE ANNOTATION PROMPT

### E.1  SLF5K EVALUATION PROMPT

The following prompt template was used to evaluate model responses on the SLF5K dataset. To prevent position bias in the evaluation, the order of model responses (analysis_1 and analysis_2) was randomly shuffled for each comparison:

Listing 4: SLF5K Evaluation Prompt

```
1   Compare and evaluate two different summaries of the same
        query. You must respond in valid JSON format.
2
3   Original Query:
4   {query}
5
6   {analysis_1_label}:
7   {response_1}
8
9   {analysis_2_label}:
10  {response_2}
11
12  Evaluation Criteria:
13  1. Accuracy (0-10):
14     - Does it capture the main points correctly?
15     - Is it faithful to the original content?
16     - Are there any factual errors?
17
```

```
18  2. Completeness (0-10):
19     - Are all key points included?
20     - Is any important information missing?
21     - Does it cover the core message?
22
23  3. Conciseness (0-10):
24     - Is it clear and to the point?
25     - Does it avoid unnecessary details?
26     - Is the language efficient?
27
28  4. Coherence (0-10):
29     - Is the summary well-organized?
30     - Does it flow logically?
31     - Is it easy to understand?
32
33  Compare both summaries and evaluate them. Respond ONLY with
        a JSON object in this exact format:
34  {
35      "{score_key_1}": {
36          "strengths": ["specific strength 1", "specific
                strength 2", ...],
37          "weaknesses": ["specific weakness 1", "specific
                weakness 2", ...]
38          "score": <overall score between 0-10>,
39          "accuracy": <score between 0-10>,
40          "completeness": <score between 0-10>,
41          "conciseness": <score between 0-10>,
42          "coherence": <score between 0-10>,
43      },
44      "{score_key_2}": {
45          "strengths": ["specific strength 1", "specific
                strength 2", ...],
46          "weaknesses": ["specific weakness 1", "specific
                weakness 2", ...]
47          "score": <overall score between 0-10>,
48          "accuracy": <score between 0-10>,
49          "completeness": <score between 0-10>,
50          "conciseness": <score between 0-10>,
51          "coherence": <score between 0-10>,
52      }
53  }
```

## E.2   KODCODE EVALUATION PROMPT

The following prompt template was used to evaluate the quality of code span selections for the KodCode dataset, which resulted in the win-rate metrics (72.17 : 7.01 : 20.82) comparing CoT feedback quality:

### Listing 5: KodCode Evaluation Prompt

```
1  Evaluate the precision and specificity of code span
       selections in two different analyses.
2
3  Problem:
4  {problem}
```

```
5
6  Solution Code:
7  {solution}
8
9  {analysis_1_label}:
10 Selected spans: {spans_1}
11 Suggestions: {improve_1}
12
13 {analysis_2_label}:
14 Selected spans: {spans_2}
15 Suggestions: {improve_2}
16
17 Please evaluate the quality of span selections in JSON
       format, focusing on precision and minimality:
18 {
19     "{score_key_1}": {
20         "score": (0-10 score for span selection precision),
21         "Reason": "Explain the reason for the score"
22     },
23     "{score_key_2}": {
24         "score": (0-10 score for span selection precision),
25         "Reason": "Explain the reason for the score"
26     },
27     "comparison": "Explain which analysis has more precise
           and minimal span selections"
28 }
29
30 Guidelines for span evaluation:
31 1. Each span should capture ONLY the specific problematic
       code, nothing more
32 2. General or overly broad selections (like entire
       functions) are penalized
33 3. Spans should not include irrelevant surrounding code
34 4. Multiple small precise spans are better than one large
       span
35 5. Spans must directly relate to the identified issue
```

### E.3 ULTRAFEEDBACK EVALUATION PROMPT

The following prompt template was used to evaluate the precision and specificity of text span selections for the UltraFeedback dataset:

Listing 6: UltraFeedback Evaluation Prompt

```
1  """Evaluate the precision and specificity of text span
       selections in two different analyses.
2
3  Problem:
4  {problem}
5
6  Response:
7  {solution}
8
9  {analysis_1_label}:
10 Problematic spans: {spans_1}
```

```
11  Good spans: {improve_1}
12
13  {analysis_2_label}:
14  Problematic spans: {spans_2}
15  Good spans: {improve_2}
16
17  Please evaluate the quality of span selections in JSON
        format, focusing on precision and minimality:
18  {
19      "{score_key_1}": {
20          "score": (0-10 score for span selection precision),
21          "Reason": "Explain the reason for the score"
22      },
23      "{score_key_2}": {
24          "score": (0-10 score for span selection precision),
25          "Reason": "Explain the reason for the score"
26      },
27      "comparison": "Explain which analysis has more precise
            span selections while avoiding unnecessary or overly
            broad spans"
28  }
29
30  Guidelines for span evaluation:
31  1. Each problematic span should capture ONLY the specific
        problematic text, nothing more
32  2. General or overly broad selections are penalized
33  3. Spans should not include irrelevant surrounding text
34  4. Multiple small precise spans are better than one large
         span
35  5. Spans must directly relate to the identified issue
36  6. Good spans should highlight the best parts of the response
```

# F  CASE STUDIES ON HUMANEVAL

We present three case studies from the HumanEval benchmark to demonstrate the effectiveness of our approach.

## F.1  SPECIAL FACTORIAL

**Problem Statement**

```python
def special_factorial(n):
    """The Brazilian factorial is defined as:
    brazilian_factorial(n) = n! * (n-1)! * (n-2)! * ... * 1!
    where n > 0

    For example:
    >>> special_factorial(4)
    288

    The function will receive an integer as input and should
        return the special
    factorial of this integer.
    """
```

**Instruct Model Solution**

```python
def special_factorial(n):
    if n < 0:
        raise ValueError("Input must be a non-negative
            integer")

    result = 1
    for i in range(1, n + 1):
        result *= math.factorial(i)

    return result
```

**PPO Model Solution**

```python
def special_factorial(n):
    result = 1
    for i in range(1, n + 1):
        result *= i
    return result
```

**Our Approach**

```python
import math
def special_factorial(n):
    if not isinstance(n, int) or n <= 0:
        raise ValueError("Input must be a positive integer.")

    result = 1
    for i in range(1, n + 1):
        result *= math.factorial(i)

    return result
```

## F.2 FILE NAME VALIDATION

---

**Problem Statement**

```python
def file_name_check(file_name):
    """Create a function which takes a string representing a
        file's name, and returns
    'Yes' if the the file's name is valid, and returns 'No'
        otherwise.
    A file's name is considered to be valid if and only if
        all the following conditions
    are met:
    - There should not be more than three digits ('0'-'9')
        in the file's name.
    - The file's name contains exactly one dot '.'
    - The substring before the dot should not be empty, and
        it starts with a letter from
    the latin alphabet ('a'-'z' and 'A'-'Z').
    - The substring after the dot should be one of these:
        ['txt', 'exe', 'dll']
    Examples:
    file_name_check("example.txt") # => 'Yes'
    file_name_check("1example.dll") # => 'No' (the name
        should start with a latin alphapet letter)
    """
```

---

**Instruct Model Solution**

```python
def file_name_check(file_name):
    # First, split the file's name by the dot
    name, extension = file_name.rsplit('.', 1)

    # Check if the extension is valid
    valid_extensions = ['txt', 'exe', 'dll']
    if extension not in valid_extensions:
        return 'No'

    # Check if the substring before the dot is not empty
    if not name:
        return 'No'

    # Check if the substring before the dot starts with a
        letter from the latin alphabet
    if not name[0].isalpha():
        return 'No'

    # Check if there are more than three digits in the
        file's name
    if sum(c.isdigit() for c in file_name) > 3:
        return 'No'

    # If all conditions are met, return 'Yes'
    return 'Yes'
```

**PPO Model Solution**

```python
def file_name_check(file_name):
    # Split the file name into two parts: the name and the
        extension
    name, extension = file_name.split('.')

    # Check if the name starts with a latin alphapet letter
    if not name[0].isalpha():
        return 'No'

    # Check if there are more than three digits in the name
    if sum(c.isdigit() for c in name) > 3:
        return 'No'

    # Check if the extension is valid
    if extension not in ['txt', 'exe', 'dll']:
        return 'No'

    # Check if the file name contains exactly one dot
    if file_name.count('.') != 1:
        return 'No'

    # If all checks pass, return 'Yes'
    return 'Yes'
```

**Our Approach**

```python
def file_name_check(file_name):
    """
    Checks if a file name is valid according to the given
        conditions.

    A file's name is considered to be valid if and only if
        all the following conditions
    are met:
    - There should not be more than three digits ('0'-'9')
        in the file's name.
    - The file's name contains exactly one dot '.'.
    - The substring before the dot should not be empty, and
        it starts with a letter from
    the latin alphabet ('a'-'z' and 'A'-'Z').
    - The substring after the dot should be one of these:
        ['txt', 'exe', 'dll'].

    Args:
        file_name (str): The name of the file to check.

    Returns:
        str: 'Yes' if the file name is valid, 'No' otherwise.
    """

    # Check if the file name contains exactly one dot
    if file_name.count('.') != 1:
        return 'No'
```

```
24      # Split the file name into two parts: before and after
            the dot
25      before_dot, after_dot = file_name.split('.')
26
27      # Check if the substring before the dot is not empty and
            starts with a letter
28      if not before_dot or not before_dot[0].isalpha():
29          return 'No'
30
31      # Check if the substring before the dot contains more
            than three digits
32      if sum(c.isdigit() for c in before_dot) > 3:
33          return 'No'
34
35      # Check if the substring after the dot is one of the
            allowed extensions
36      if after_dot not in ['txt', 'exe', 'dll']:
37          return 'No'
38
39      # If all conditions are met, return 'Yes'
40      return 'Yes'
```

## G  LIMITATIONS

Despite its effectiveness, TEXT2GRAD has two potential limitations. First, our framework still depends on the quality of the reward model. While this dependence is inherent to RLHF-style methods, our CoT-based annotation pipeline achieves strong human alignment (82–94% accuracy across datasets, Table 16) and maintains high grounding fidelity with unmatched-span rates below 2.5% (Table 15). The framework's consistent performance gains across all benchmarks demonstrate that the current reward model quality is sufficient for effective policy learning, though further improvements in critique generation could enhance optimization in tasks requiring highly nuanced feedback.

Second, generating and applying token-level rewards introduces additional computational overhead compared to scalar reward methods. Our analysis (Appendix K) shows that this overhead is modest: approximately 9–11% per training step (Table 13), primarily from a single reward-model forward pass per trajectory. Moreover, span-level annotation reduces token costs by 85–90% compared to dense token-level labeling, with annotation costs on the order of $10^{-3}$ USD per sample (Table 14). The substantial performance improvements help justify this modest cost increase, and the framework remains practical for large-scale deployments.

In future work, we aim to further improve reward model precision and efficiency, and to extend our framework to broader generation settings, including more open-ended tasks where fine-grained feedback is harder to define.

## H  TRAINING DYNAMICS ON KODCODE AND ULTRAFEEDBACK

Figure 5 compares the training dynamics of TEXT2GRAD against standard PPO on the KodCode and UltraFeedback datasets. In both cases, TEXT2GRAD exhibits significantly more stable convergence behavior, while PPO suffers from reward oscillation and inconsistent policy updates — indicative of poor gradient signal utilization in scalar-reward settings.

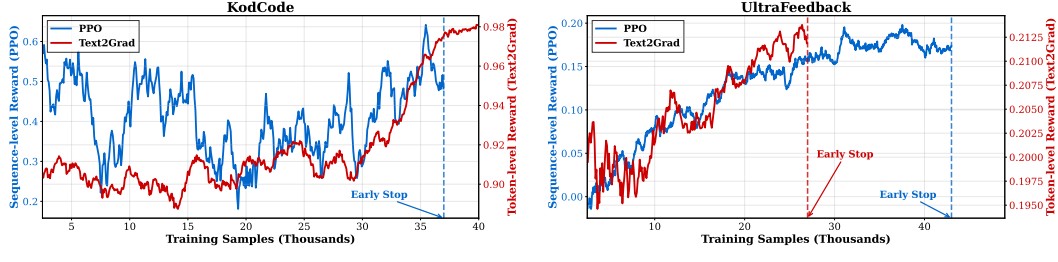

(a) KodCode: TEXT2GRAD (red) vs. PPO (blue)    (b) UltraFeedback: TEXT2GRAD (red) vs. PPO (blue)

Figure 5: Training reward curves comparing TEXT2GRAD and standard PPO. TEXT2GRAD demonstrates smoother, more consistent optimization with reduced oscillation — particularly critical in structured domains like code generation (KodCode) and nuanced preference modeling (UltraFeedback). Shaded regions (if present) indicate one standard deviation over three random seeds.

# I  ABLATION STUDY ON SLF5K

To evaluate the contribution of key components in our framework, we conduct ablation studies on the SLF5K dataset. Table 2 in the main text shows that removing Chain-of-Thought (CoT) reasoning leads to consistent performance degradation across all metrics, confirming its importance for guiding fine-grained policy updates.

Figure 6 further illustrates the training dynamics by comparing the win rate of our full model against the variant without CoT reasoning. The consistent performance gap demonstrates that CoT-enhanced natural language feedback provides more actionable and semantically grounded signals for policy optimization.

## I.1  EFFECT OF SPAN-BASED REWARD SELECTION

To directly address the core design choice in our method, we compare five reward strategies on SLF5K examining three orthogonal dimensions: supervision granularity (dense token-level vs. span-level), feedback source (human vs. model-generated), and within-span token weighting strategies (Table 11).

The dense token baseline performs substantially worse despite maximal supervision density. Analysis reveals that dense labeling produces a highly skewed distribution with approximately 70% of tokens labeled, predominantly on verbs and function words rather than semantically meaningful spans. This introduces noise into advantage estimates and destabilizes training, while costing 6–8× more in annotation tokens.

For within-span token weighting, we test two alternative strategies against our uniform assignment. **Token Importance** assigns full rewards ($\pm 1.0$) only to nouns and verbs within each span, while assigning reduced weights ($\pm 0.2$) to others, under the hypothesis that content words carry greater semantic responsibility. **Linear Decay** assigns rewards proportionally to token position: for a span of length $n$, the $i$-th token receives $r_i = 1.0 - (i-1) \cdot \frac{0.9}{n-1}$ for positive spans (and negated for negative spans), concentrating credit on early tokens under the assumption that initial errors dominate in autoregressive generation.

As shown in Table 11, both alternative weighting schemes underperform uniform assignment. Token Importance introduces part-of-speech classification noise that disrupts span coherence, while Linear Decay amplifies variance in the advantage estimator by concentrating gradients on initial tokens. Using human feedback with our span-to-token mapping achieves intermediate performance, confirming that span selection itself provides value. The full TEXT2GRAD method with CoT critiques and uniform weighting achieves the best results, demonstrating that balanced credit assignment combined with structured reasoning sharpens span precision beyond natural human annotation and gradient reweighting heuristics. Our span-based approach maintains high grounding fidelity (unmatched rate <2.5%) while producing more stable advantages at substantially lower cost.

Table 11: Ablation study on reward design choices (SLF5K). **Bold**: best results.

| Method | R-L | BLEU | BERT-F1 |
|---|---|---|---|
| *Reward Granularity* | | | |
|     Dense Token Reward | 0.196 | 0.022 | 0.888 |
| *Token Weighting Strategy* | | | |
|     Token Importance | 0.237 | 0.059 | 0.889 |
|     Linear Decay | 0.253 | 0.068 | 0.890 |
| *Feedback Source* | | | |
|     Human Feedback (Uniform) | 0.286 | 0.091 | 0.900 |
| *Ours (Full)* | | | |
|     GPT-4o + CoT (Uniform) | **0.291** | **0.094** | **0.902** |

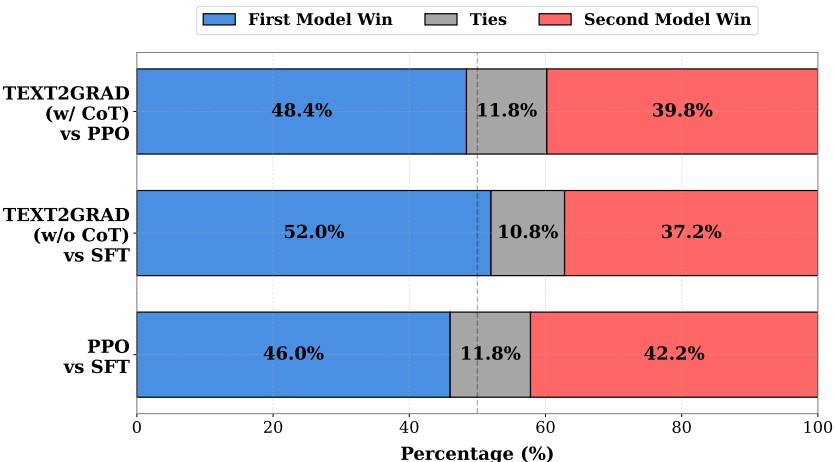

Figure 6: Win rate comparison during training on SLF5K. Our full model (TEXT2GRAD-8B with CoT) consistently outperforms the ablated variant (without CoT), validating the effectiveness of structured reasoning in generating high-quality natural language rewards.

# J PSEUDOCODE FOR THE TEXT2GRAD FRAMEWORK

---

**Algorithm 1** TEXT2GRAD: Reinforcement Learning from Natural Language Feedback (Overall Framework)

---

    **Input:** Set of prompts for policy training.
    **Output:** Optimized policy $\pi_\theta$.

    **Phase 1: Dual-Feedback Reward Annotation (Described in Section 3.3)**
1: Initialize dataset for reward model training $\mathcal{D}_R \leftarrow \emptyset$.
2: Generate initial responses $y_i$ for a set of prompts $x_i$ (e.g., using a base policy).
3: **for all** prompt $x_i$ and its corresponding response $y_i$ **do**
4:     $(c_i, \mathcal{A}(y_i), \boldsymbol{\delta}_i) \leftarrow$ GenerateDualFeedback$(x_i, y_i)$           $\triangleright$ See Algorithm 2
5:     Let $z_i \leftarrow [c_i; \mathcal{A}(y_i)]$          $\triangleright$ $c_i$ is critique, $\mathcal{A}(y_i)$ is span-JSON
6:     Add $(x_i, y_i, z_i)$ to $\mathcal{D}_R$.
7: **end for**

    **Phase 2: Reward Model Training (Described in Section 3.4)**
8: $R_\phi \leftarrow$ TrainRewardModel$(\mathcal{D}_R)$           $\triangleright$ See Algorithm 3

    **Phase 3: NL-Gradient Policy Optimization (Described in Section 3.5)**
9: Initialize policy $\pi_\theta$ (e.g., with a pre-trained LLM) and value function $V_\psi$.
10: $\pi_\theta \leftarrow$ OptimizePolicyWithNLGradient$(\pi_\theta, R_\phi, V_\psi)$          $\triangleright$ See Algorithm 4
11: **return** Optimized policy $\pi_\theta$.

---

---

**Algorithm 2** Dual-Feedback Reward Annotation (Section 3.3)

---

1: **procedure** GENERATEDUALFEEDBACK($x, y$)
2:     **Input:** Prompt $x$, generated response $y = (y_1, \ldots, y_T)$.
3:     **Output:** Natural language critique $c$, structured span-level reward map $\mathcal{A}(y)$, token-level pseudo-rewards $\boldsymbol{\delta}$.
    *// Dual-Feedback Annotation using a strong LLM (e.g., GPT-4o)*
4:     **if** human-written feedback is lacking (Reasoning-Augmented Annotation) **then**
5:         Guide LLM to:
6:             (1) Reason about the quality of response $y$ step-by-step.
7:             (2) Output a critique $c$ based on this reasoning.
8:             (3) Produce a span-level JSON map $\mathcal{A}(y)$ associating spans $s_k \subset y$ with labels $\ell_k \in \{\texttt{positive}, \texttt{neutral}, \texttt{negative}\}$.
9:     **else**
10:         Prompt LLM to output critique $c$ and span-level JSON map $\mathcal{A}(y)$.
11:     **end if**
12:                                   $\triangleright$ Formally, $R_{\text{LLM}}(x, y) = (c, \mathcal{A}(y))$, where $\mathcal{A}(y) : s_k \mapsto \ell_k$
    *// Token-Level Reward Mapping*
13:     Initialize token-level pseudo-rewards $\boldsymbol{\delta} = (\delta_1, \ldots, \delta_T)$ with zeros.
14:     **for all** labeled span $s_k$ in $\mathcal{A}(y)$ **do**
15:         Let $\ell_k = \mathcal{A}(y)[s_k]$.
16:         **if** $\ell_k = \texttt{positive}$ **then**
17:             **for all** token index $t$ such that $y_t \in s_k$ **do**
18:                 $\delta_t \leftarrow +1$.
19:             **end for**
20:         **else if** $\ell_k = \texttt{negative}$ **then**
21:             **for all** token index $t$ such that $y_t \in s_k$ **do**
22:                 $\delta_t \leftarrow -1$.
23:             **end for**
24:         **end if**         $\triangleright$ `neutral` spans are typically unannotated and default to $\delta_t = 0$.
25:     **end for**
26:     **return** $c, \mathcal{A}(y), \boldsymbol{\delta}$.
27: **end procedure**

---

**Algorithm 3** Reward Model Training (Section 3.4)

---

1: **procedure** TRAINREWARDMODEL($\mathcal{D}_R$)
2:     **Input:** Dataset $\mathcal{D}_R = \{(x_i, y_i, z_i)\}_{i=1}^N$, where $z_i = [c_i; \mathcal{A}(y_i)]$.
3:     **Output:** Trained reward model $R_\phi$.
4:     Initialize reward model parameters $\phi$.
5:     The reward model $R_\phi$ is trained to predict $z$ given $x, y$: $p_\phi(z \mid x, y) = \prod_{j=1}^{|z|} p_\phi(z_j \mid z_{<j}, x, y)$.
6:     Define the loss function: $\mathcal{L}_R(\phi) = -\mathbb{E}_{(x,y,z) \in \mathcal{D}_R} [\log p_\phi(z \mid x, y)]$.
7:     Train $R_\phi$ by minimizing $\mathcal{L}_R(\phi)$ on $\mathcal{D}_R$ using teacher forcing and a standard causal LM objective.
8:     **return** Trained reward model $R_\phi$.
9: **end procedure**

---

---

**Algorithm 4** NL-Gradient Policy Optimization (Section 3.5)

---

1: **procedure** OPTIMIZEPOLICYWITHNLGRADIENT($\pi_{\theta_{\text{init}}}, R_\phi, V_{\psi_{\text{init}}}$)
2:     **Input:** Initial policy $\pi_{\theta_{\text{init}}}$, trained reward model $R_\phi$, initial value function $V_{\psi_{\text{init}}}$.
3:     **Hyperparameters:** Learning rates, PPO clipping $\epsilon$, entropy bonus $\beta$, GAE $\gamma, \lambda$.
4:     **Output:** Optimized policy $\pi_\theta$.
5:     Initialize policy $\pi_\theta \leftarrow \pi_{\theta_{\text{init}}}$, value function $V_\psi \leftarrow V_{\psi_{\text{init}}}$.
6:     **for** each iteration $iter = 1, \ldots, \text{MaxIterations}$ **do**
7:         Let $\pi_{\theta_{\text{old}}} \leftarrow \pi_\theta$.
8:         Initialize a batch of rollouts $\mathcal{B} \leftarrow \emptyset$.
9:         **for** each sample $s = 1, \ldots, \text{NumSamplesPerIteration}$ **do**
10:             Sample prompt $x$.
11:             Generate response $y = (y_1, \ldots, y_T) \sim \pi_{\theta_{\text{old}}}(\cdot \mid x)$.
12:             Generate feedback $z' = [c'; \mathcal{A}'(y)] \sim R_\phi(z' \mid x, y)$.
13:             Parse $\mathcal{A}'(y)$ to get token-level pseudo-rewards $\boldsymbol{\delta}' = (\delta_1', \ldots, \delta_T')$ (using lines 11-20 of Alg. 2).
14:             For $t = 1, \ldots, T$: $r_t^{\text{total},A} \leftarrow \delta_t' + r_t^{\text{KL}}$      $\triangleright r_t^{\text{KL}}$ is an optional KL-penalty term.
15:             Compute advantages $A_1, \ldots, A_T$. For $t = T \ldots 1$:
16:             $A_t = \sum_{k=t}^{T} \gamma^{k-t} r_k^{\text{total},A} - V_\psi(x, y_{<t})$.    (Or use GAE: $A_t = \sum_{l=0}^{T-t-1} (\gamma\lambda)^l (r_{t+l}^{\text{total},A} + \gamma V_\psi(x, y_{<t+l+1}) - V_\psi(x, y_{<t+l}))$ )
17:             Add $(x, y, \boldsymbol{\delta}', \mathbf{A}, \mathbf{r}^{\text{total},A})$ to $\mathcal{B}$.
18:         **end for**
19:         **for** each epoch $e = 1, \ldots, \text{NumEpochsPPO}$ **do**
20:             **for all** $(x, y, \boldsymbol{\delta}', \mathbf{A}, \mathbf{r}^{\text{total},A})$ in $\mathcal{B}$ **do**
21:                 For $t = 1, \ldots, T$:
22:                 $\rho_t(\theta) = \frac{\pi_\theta(y_t|x,y_{<t})}{\pi_{\theta_{\text{old}}}(y_t|x,y_{<t})}$.
23:                 $L_t^{\text{CLIP}}(\theta) = \min\left(\rho_t(\theta)A_t, \; \text{clip}(\rho_t(\theta), 1-\epsilon, 1+\epsilon)A_t\right)$.
24:                 $L_t^{\text{VF}}(\psi) = (V_\psi(x, y_{<t}) - (\sum_{k=t}^{T} \gamma^{k-t} r_k^{\text{total},A}))^2$.     $\triangleright$ Value target is discounted sum of rewards.
25:                 $L_t^{\text{ENT}}(\theta) = \mathcal{H}(\pi_\theta(\cdot \mid x, y_{<t}))$.
26:             **end for**
27:             $L^{\text{PPO}}(\theta) = \mathbb{E}_{\mathcal{B},t}\left[L_t^{\text{CLIP}}(\theta) - \beta L_t^{\text{ENT}}(\theta)\right]$.
28:             $L^{\text{VF}}(\psi) = \mathbb{E}_{\mathcal{B},t}\left[L_t^{\text{VF}}(\psi)\right]$.
29:             Update policy parameters: $\theta \leftarrow \text{optimizer\_step}(\theta, \nabla_\theta L^{\text{PPO}}(\theta))$.
30:             Update value function parameters: $\psi \leftarrow \text{optimizer\_step}(\psi, \nabla_\psi L^{\text{VF}}(\psi))$.
31:         **end for**
32:     **end for**
33:     **return** Optimized policy $\pi_\theta$.
34: **end procedure**

---

## K  ANNOTATION AND TRAINING EFFICIENCY ANALYSIS

Table 12 compares the token consumption between span-level and token-level annotation approaches across our experimental datasets.

Table 12: Annotation efficiency: Token-level vs. Span-level comparison.

| Dataset | Method | Tokens/Sample | Total Tokens |
|---------|--------|---------------|--------------|
| UltraFeedback | Token-Level | 273.3 | 16.7M |
|  | Span-Level | 40.1 | 2.4M (**85%↓**) |
| KodCode | Token-Level | 170.4 | 1.5M |
|  | Span-Level | 50.1 | 0.5M (**70%↓**) |

First, we report wall-clock training time under identical hardware, data splits, and parallelism settings. Table 13 compares PPO and TEXT2GRAD using the same GPU configuration (detailed in Appendix D).

Table 13: Wall-clock training time comparison (minutes per step).

| Dataset | PPO | TEXT2GRAD | Overhead |
|---------|-----|-----------|----------|
| KodCode | 0.54 | 0.60 | +11% |
| UltraFeedback | 0.88 | 0.96 | +9% |
| SLF5K | 0.44 | 0.49 | +11% |

The extra time comes almost entirely from one additional forward pass of the reward model per sampled trajectory. This single autoregressive pass jointly produces both the critique and the span map; it does not add extra decoding stages or any backpropagation through the reward model.

Second, we quantify the annotation efficiency of the span-based pipeline. As shown in Table 14, the total annotation budget is small, and the cost per training sample remains on the order of $10^{-3}$ USD, which is consistent with the near-parity in training time versus PPO.

Table 14: Span-level annotation cost breakdown (GPT-4o pricing).

| Dataset | Input Tokens | Output Tokens | Total (USD) | Per Sample |
|---------|--------------|---------------|-------------|------------|
| UltraFeedback | 5.18M | 1.21M | $44.06 | $0.0036 |
| SLF5K | 1.23M | 0.12M | $7.99 | $0.0016 |
| KodCode | 2.74M | 0.46M | $20.60 | $0.0025 |
| **Total** | **9.15M** | **1.79M** | **$72.65** | – |

## L  SPAN GENERATION FIDELITY ANALYSIS

To ensure the reliability of our reward signals, we verify that all generated spans are exact quotes from the model responses. Our annotation pipeline includes explicit instructions requiring "spans must be exact quotes from the response" and automated post-processing to remove any unmatched cases.

Table 15: Span generation fidelity: Unmatched span rates across datasets.

| Dataset | Unmatched Rate | Fidelity |
|---------|----------------|----------|
| KodCode | 0.93% | **99.07%** |
| SLF5K | 1.30% | 98.70% |
| UltraFeedback | 2.47% | 97.53% |
| *Average* | *1.57%* | *98.43%* |

The consistently low proportion of unmatched cases (under 2.5%) demonstrates the high fidelity of our span generation process, ensuring that reward signals are grounded in actual model outputs rather than fabricated or paraphrased content.

**Robustness to Tokenization Mismatch.** Our pipeline is explicitly designed to handle tokenizer differences between the annotation model (GPT-4o) and policy model (LLaMA). GPT-4o generates spans as character-level substrings (exact quotes), not token sequences, decoupling span identification from any specific tokenizer. The mapping procedure operates in three steps: (1) GPT-4o identifies spans as exact character-level quotes from the original response, (2) we locate the character interval [start, end] of each span in the raw text, and (3) we re-tokenize this character interval using the policy model's tokenizer to obtain token indices. Reward attribution and policy updates operate entirely within the policy model's token space, eliminating dependency on GPT-4o's tokenization.

The low unmatched-span rates (0.93–2.47%) demonstrate empirical robustness. Given that only $\sim$30% of tokens receive non-zero rewards, these error rates have minimal impact on gradient quality. When boundary mismatches occur due to different byte-pair merges, the affected regions default to zero reward, preserving gradient safety without introducing spurious signals.

## M    HUMAN ALIGNMENT ANALYSIS

To validate the quality of our reward model's critique and span-level annotations, we conducted a comprehensive human evaluation study across all three datasets. For each dataset, we randomly sampled 100 instances and recruited three human annotators with expertise in the respective domains to evaluate the quality of generated critiques and their corresponding span selections.

Human annotators were asked to evaluate two aspects of our reward model's output:

**Critique Quality:** Assess whether the natural language critique accurately identifies the strengths and weaknesses of the model response.

**Span Alignment:** Evaluate whether the selected spans (both positive and negative) are correctly identified and properly justified by the critique, using a binary scale (Correct/Incorrect).

Table 16: Human evaluation: Agreement with reward model annotations.

| Dataset | Accuracy | Task Type |
|---------|----------|-----------|
| KodCode | **94%** | Code (Objective) |
| SLF5K | 86% | Summarization |
| UltraFeedback | 82% | Open-domain QA |

As shown in Table 16, KodCode achieves the highest human agreement (94%), probably due to its well-defined code-centric critique tasks with objective ground truths. SLF5K exhibits strong alignment (86%), reflecting moderate subjectivity in general text evaluation. In contrast, UltraFeedback shows the lowest accuracy (82%), which we attribute to its open-ended, reasoning-heavy nature, where human annotators exhibit greater interrater variability due to the lack of rigid criteria. This trend confirms that our reward model performs most reliably in structured domains and remains robust even under high subjectivity, validating its practical utility across diverse evaluation paradigms.

## M.1 Error Handling and Failure Mode Analysis

To address concerns about incorrect or adversarial feedback, we analyzed failure modes in GPT-4o-generated annotations. Incorrect annotations are rare ($<3\%$) and primarily involve loosely grounded or overly broad spans rather than hallucinated critiques.

We mitigate these errors through two mechanisms: (1) **CoT-based reasoning prompts** (Section 3.3), which enforce explicit justification before span selection, ensuring that every labeled span must be anchored to evidence in the critique; and (2) **Offline span-validation pass**, which filters or re-annotates inconsistent annotations before reward-model training. This validation step checks for exact-quote matching (Table 15) and removes cases where spans cannot be found in the original response.

Consequently, residual noise is minimal and has no measurable effect on downstream policy optimization. The combination of high human agreement (82–94%), low unmatched-span rates ($<2.5\%$), and validation safeguards demonstrates that our annotation pipeline is empirically high-quality and robust to occasional GPT-4o errors.

## N Baseline Settings: PRM-PPO, DPO, and PPO

For a fair comparison, we followed the methodology of Lightman et al. (2023), which defines Process Reward Models (PRMs) through explicit step-level supervision. This section details how PRM spans or steps were defined in each domain and explains why PRM-PPO was excluded on UltraFeedback.

**Definition of PRM Spans or Steps.** On SLF5K (summarization), GPT-4o decomposed each response into sentence- or clause-level content units, assigning binary correctness labels based on factual alignment with the reference summary. On KodCode (code generation), GPT-4o segmented each program into code blocks or logical statements and labeled each according to unit-test outcomes or reference execution traces. These labeled spans served as the oracle for reward-model training and subsequent PPO optimization.

Table 17: Step-level annotations and PRM F1 scores for SLF5K and KodCode.

| Dataset | PRM F1 (%) | Annotation Note |
|---------|-----------|-----------------|
| SLF5K | 78.17 | Sentence-/clause-level factual correctness based on alignment with the reference summary |
| KodCode | 80.63 | Code-block or logical-statement correctness based on unit-test and execution-trace outcomes |

**Exclusion on UltraFeedback.** We did not include PRM-PPO on UltraFeedback because the dataset represents open-domain QA, where responses are long ($\approx 276$ tokens on average) and lack clear intermediate reasoning steps. As Lightman et al. (2023) show, PRMs are most effective in tasks with explicit multi-step reasoning, such as mathematics and code, where intermediate verification is possible. In contrast, **?** survey that PRMs are primarily applied to structured reasoning domains—math, programming, multimodal reasoning, and robotics—while outcome-level or semantic-feedback models remain more appropriate for general QA.

Applying PRM supervision to UltraFeedback would thus be both conceptually unsuitable and computationally expensive. Each PRM annotation would require GPT-4o to decompose the full answer into reasoning steps and verify each step's correctness, resulting in a 6–8$\times$ higher token budget compared with our single-pass critique $\rightarrow$ span annotation pipeline. Given the absence of explicit reasoning chains and the high annotation cost, we excluded PRM-PPO for this domain.

**Reward Signals for DPO and PPO Baselines.** For SLF5K and UltraFeedback, we use the datasets' existing chosen–rejected pairs and scalar preferences. DPO is trained exactly as in the original formulation using these preference pairs. PPO uses a scalar-valued reward model trained on the same preference data via pairwise ranking.

For KodCode, where step-level correctness is directly verifiable, we construct chosen/rejected labels automatically: each candidate program is executed against unit tests, and passing vs. failing runs

define preferences. We additionally include outputs from GPT-4o and DeepSeek-r1 to ensure diverse candidates. These labels are then fed into both DPO and PPO to ensure a fair comparison with TEXT2GRAD.

## O    SPAN LENGTH ANALYSIS

Our method does not impose fixed span lengths; spans are generated dynamically by the reward model based on response content and structure. To analyze how CoT reasoning affects span length selection and annotation quality, we examine the span length distribution and the relationship between span length and annotation accuracy on SLF5K.

Table 18 shows the span length distribution statistics, comparing CoT-enabled and NoCoT variants. The results reveal that CoT reasoning produces slightly shorter Good spans but is more willing to precisely localize problematic regions in Poor spans. This suggests that CoT reasoning enables more targeted feedback by avoiding overly broad span selections. Q1, Q2, and Q3 represent the 25th-, 50th-, and 75th-percentile average token lengths per span, respectively.

Table 18: Span length distribution statistics on SLF5K (tokens per span).

| Metric | Good Spans | | Poor Spans | |
|---|---|---|---|---|
| | CoT | NoCoT | CoT | NoCoT |
| Mean | 8.59 | 8.09 | 6.94 | 7.28 |
| Median | 7.5 | 7.0 | 6.0 | 7.0 |
| Min | 2 | 2 | 1 | 1 |
| Max | 27 | 29 | 22 | 18 |
| Q1 (25%) | 5 | 5 | 4 | 5 |
| Q2 (50%) | 7.5 | 7.0 | 6.0 | 7.0 |
| Q3 (75%) | 11 | 10 | 9 | 9 |

We further quantify whether span length affects annotation quality by measuring accuracy across different length buckets. Table 19 shows span annotation accuracy stratified by quartiles of span length. Critically, with CoT reasoning, annotation accuracy remains consistently high (93.1%–96.4%) across all length ranges for both Good and Poor spans, demonstrating robustness to span length variation. In contrast, without CoT, we observe boundary effects: accuracy is notably higher in the Q1–Q2 range (94.1%–94.9%) but drops in shorter (0–Q1) and longer (Q4–100%) spans, suggesting that NoCoT struggles with both very short and very long span selections.

Table 19: Span annotation accuracy (%) by length quartile on SLF5K.

| Length Range | Good Spans | | Poor Spans | |
|---|---|---|---|---|
| | NoCoT | CoT | NoCoT | CoT |
| 0–Q1 | 89.8 | **96.1** | 93.3 | **96.4** |
| Q1–Q2 | 94.1 | 93.4 | 94.9 | **96.2** |
| Q2–Q3 | 90.5 | **95.6** | 92.0 | 94.7 |
| Q3–100% | 91.2 | 93.1 | 91.4 | **95.3** |
| *Avg.* | *91.4* | ***94.6*** | *92.9* | ***95.7*** |

This length-invariant accuracy with CoT reasoning directly improves downstream performance. As shown in Table 2, Text2Grad with CoT achieves ROUGE-L 0.291 and BERTScore 0.902, outperforming the NoCoT variant (0.275 and 0.898, respectively). This performance gap confirms that inaccurate or noisy span selections—particularly the boundary effects observed without CoT—degrade policy learning effectiveness. CoT reasoning enables dynamic, content-driven span selection that maintains 93–96% annotation accuracy across all lengths, providing clean signals for policy gradients regardless of granularity.

## P    LLM USAGE STATEMENT

Large Language Models (LLMs) played a significant role in both the research methodology and writing phases of this work. We acknowledge that their contribution was substantial enough to warrant detailed disclosure as per ICLR guidelines.

**Research Methodology:** We used OpenAI's GPT-4o model as a core component of our data generation pipeline. Specifically, GPT-4o was employed to generate the dual-feedback annotations (textual critiques and span-level labels) that constitute the training data for our reward model, as detailed in Section 3.3. This automated annotation process was essential for scaling our approach and represents a fundamental aspect of our methodology. The LLM was prompted with carefully designed instructions to ensure consistency and quality of the generated annotations.

**Writing Assistance:** We also used LLMs for language polishing and proofreading of this paper to improve clarity and readability. However, all technical content, experimental design, analysis, and conclusions were developed by the human authors.

