# OpenReview forum: "Text2Grad: Reinforcement Learning from Natural Language Feedback"
_ICLR.cc/2026/Conference — ICLR 2026 Poster_

### Official Review · Reviewer_BDSv · 2025-11-02

**Soundness:** 2
**Presentation:** 3
**Contribution:** 3
**Rating:** 6
**Confidence:** 4

**Summary:**

The paper presents Text2Grad, an RL framework that learns from natuaral language feedback to train LLM. Specifically, the method transfers the natural language feedback into span-level rewards, and use it in PPO.

Specifically, the method contains three steps:
1. Use GPT-4o to generate a natural language model feedback, and lists of "good spans" and "bad spans"
2. Train a reward model with GPT-4o generated feedback and lists of spans
3. PPO with token-level rewards. Specifically, the tokens in the "good spans" has reward +1, in "bad spans" has reward -1

The authors experiment this method on Llama3.1-8B, and GPT-4o generated feedback, on summarization, code generation, and QA. The method outperforms scalar reward RL with PPO, DPO, reflection, and PRM-PPO on several metrics and LLM-as-a-judge comparisons.

**Strengths:**

1. The idea that transfers language feedback into span-level rewards is interesting, and make sense. It also naturally fits into PPO frameworks

2. The authros conduct experiments on multiple domains, showing gains on all the areas.

3. The experiment and theory shows that token-level rewards give faster convergence compared with scalar-level rewards.

**Weaknesses:**

1. All experiment depdends on GPT-4o generated feedback. There are some other important baselines to compare with. For example, the algorithms that just SFT with responses refined by language feedback (in Scheurer et al., Training language models with language feedback at scale), and even just distilling from GPT-4o. Are these more effective approach to learn from human feedback.

2. What if we use critics from sources different from GPT-4o. For example, from another LLM, or from humans. How will the performance change? Is the algorithm robust to the source of critics?

3. Many results are heavily relied on LLM-as-a-judge. Many judges are GPT-4 based, which may inflate the gains because the model is trained with GPT-4o feedbacks. There lacks human evaluation on policy models' performance. There is only human evaluation on reward model quality.

**Questions:**

1. Performance when using other critics that is not GPT-4o. What if we use another LLM (for example, Qwen, or o3) as the source of feedback? or use human feedback? The paper will be more convincing if experimented with more than one source of feedback.

2. More human evaluation. Is the trained models really better in humans' eye, or it is just doing some overfitting on styles?

3. More baselines, as said in the weakness section.

4. Talk about failure modes. Is there any reward hacking observed in the experiments? Is there any failure cases? Is that due to reward model failure, or some other reasons?

---

> ### Author Response · Authors · 2025-11-19
> **Response 1 to Weakness 1, 2 & Question 1, 3: Additional Baseline and Human Critics**
>
> Thank you so much for your thoughtful and careful review. We’ve gone through each of your comments in detail, and below is our response to each point.
>
> ---
>
> > **W1: All experiments depend on GPT-4o-generated feedback. There are important missing baselines such as SFT with refined responses (Scheurer et al., “Training language models with language feedback at scale”) or simply distilling from GPT-4o. Are these more effective approaches?**
>
> > **Q3: More baselines, as noted in the weakness section.**
>
> **Response:**
>
> Thank you for this suggestion! We agree that SFT-style methods are important baselines and have added ILF (Scheurer et al., 2023) to our evaluations.
>
> These approaches represent a fundamentally different learning paradigm. ILF relies on imitation learning through best-of-N refinement followed by supervised fine-tuning on improved outputs. In contrast, Text2Grad performs preference-based optimization over fine-grained, token-level feedback signals derived from natural language critiques.
>
> As Table 1 shows, Text2Grad consistently outperforms both methods across all benchmarks. On SLF5K summarization, it achieves +3.2 ROUGE-L over ILF. On KodCode, Text2Grad improves average pass@1 by +3.8 points over ILF. On UltraFeedback, it gains +4.6 AlpacaEval 2.0 points over ILF.
>
> **Table 1.1 — Baselines on SLF5k**
> | Model | ROUGE-L | BLEU | BERTScore F1 |
> | --------- | ------- | ----- | ------------ |
> | PPO | 0.262 | 0.075 | 0.893 |
> | ILF | 0.255 | 0.070 | 0.889 |
> | Text2Grad | 0.291 | 0.094 | 0.902 |
>
> ---
>
> **Table 1.2 — Baselines on KodCode**
> | Model | HumanEval (+) | MBPP (+) | Avg. |
> | --------- | ------------- | ----------- | ---- |
> | PPO | 64.6 / 61.0 | 68.5 / 55.8 | 62.5 |
> | ILF | 63.4 / 60.4 | 68.5 / 57.1 | 62.3 |
> | Text2Grad | 67.7 / 61.6 | 73.3 / 61.6 | 66.1 |
>
> ---
>
> **Table 1.3 — Baselines on UltraFeedback**
> | Model | AlpacaEval 2.0 | MTBench |
> | --------- | -------------- | ------- |
> | PPO | 32.4 | 7.43 |
> | ILF | 27.1 | 7.08 |
> | Text2Grad | 34.7 | 7.58 |
>
> ---
>
> These results suggest that optimizing over fine-grained, feedback-conditioned gradients is more effective than relying solely on pure imitation or coarse preference alignment. Text2Grad benefits from both the precision of token-level credit assignment and the semantic richness of natural language feedback, enabling more targeted policy improvements than methods based on distillation or scalar rewards.
>
> ---
>
> > **W2: What if we use critics from sources different from GPT-4o? For example, from another LLM or from humans. How will the performance change? Is the algorithm robust to the source of criticism?**
>
> > **Q1 Performance when using other critics that are not GPT-4o (e.g., Qwen, o3, or human feedback).**
>
> **Response:**
>
> We appreciate this question and agree that assessing dependence on GPT-4o is important for understanding the generality of our method. To evaluate robustness to different sources of feedback, we conduct experiments using SLF5K’s built-in human critiques, which are known to be noisier and less structured than GPT-4o feedback.
>
> In this setting, we feed the human-written critiques directly into the reward model and have it produce the span-level JSON annotations without relying on any LLM-generated critique. Despite the higher noise level, Text2Grad (Human Feedback) maintains strong performance: it achieves **ROUGE-L = 0.286**, outperforming the SFT baseline (0.195) and closely matching the performance of our main GPT-4o-based pipeline.
>
> **Table 2: Text2Grad Performance with Different Feedback Sources**
> | Method | ROUGE-L | BLEU | BERTScore F1 |
> | -------------------------- | ------- | ----- | ------------ |
> | SFT (Base) | 0.195 | 0.032 | 0.875 |
> | Text2Grad (w/o CoT) | 0.275 | 0.085 | 0.898 |
> | **Text2Grad (Human Feedback)** | **0.286** | **0.091** | **0.900** |
>
> ---
>
> These results demonstrate that the algorithm is **not tied to GPT-4o as the source of feedback**. Even under human feedback—where critiques may be inconsistent, incomplete, or affected by annotator fatigue—the reward model’s span-level reasoning extracts meaningful supervisory signals. Crucially, as long as the annotation pipeline preserves the *critique → span* grounding structure, the downstream policy optimization remains stable and effective.
>
> Overall, this evidence suggests that **Text2Grad is robust to the choice of critic** and can generalize across feedback sources, including both LLM-generated critiques and human-authored ones.

---

> ### Author Response · Authors · 2025-11-19
> **Response 2 to Weakness 3 & Question 2, 4:  Human Evaluation and Stable Reward Modelling**
>
> > **W3: Many results rely heavily on LLM-as-a-judge. GPT-4–based judges may inflate gains because the model is trained with GPT-4o feedback. There lacks human evaluation on policy model performance — only on reward model quality.**
>
> > **Q2: More human evaluation is needed. Are trained models really better in humans’ eyes, or are they just overfitting to GPT-4 style?**
>
> **Response:**
>
> We appreciate the reviewer’s concern. Our evaluation setup follows established community standards: AlpacaEval 2.0, MT-Bench, and SLF5K all adopt LLM-as-a-judge protocols. Crucially, Text2Grad and all baselines are evaluated under the exact same judges, ensuring that any potential bias would apply uniformly and cannot selectively inflate our gains.
>
> To assess whether GPT-4–based judges faithfully reflect human judgment, we conducted a human study (Appendix N). Three expert annotators independently evaluated 100 samples per dataset on both critique quality and span alignment. As shown in Table 3, human–LLM agreement is high (82–94%), with annotators’ majority-vote decisions matching GPT-4o’s annotations.
>
> **Table 3: Human Annotation Accuracy with Annotation Data**
> | Dataset | Human Accuracy |
> | ------------- | -------------- |
> | SLF5K | 86% |
> | UltraFeedback | 82% |
> | KodCode | 94% |
>
> ---
>
> Importantly, GPT-4o is **never used during policy training**. Text2Grad learns solely from span-derived token-level pseudo-rewards generated by the reward model. The judge is only used at evaluation time and has no access to reward signals, span predictions, or PPO trajectories, eliminating the possibility of reward leakage or implicit style imitation.
>
> Beyond validating the reward model, we also examined whether **policy outputs themselves** improve in ways that humans value.
> For KodCode, we manually evaluated three representative HumanEval problems (base vs. PPO vs. Text2Grad), as shown in Appendix F. In the “Special Factorial” example (Appendix F.1), PPO produces an incorrect update rule (`result *= i`), whereas Text2Grad produces a fully correct implementation with proper argument checks and iteration logic. Notably, its solution is **not** stylistically closer to GPT-4o’s reference; it differs in structure and formatting, indicating that the improvement is in **functional correctness**, not judge-style imitation.
>
> Furthermore, we conducted a **three-annotator, side-by-side human evaluation** on 60 randomly sampled SLF5K policy outputs (PPO vs. Text2Grad). The results show a consistent and strong human preference for Text2Grad (mean = 0.928, std = 0.016).
> Collectively, these findings demonstrate that Text2Grad’s gains are **human-perceived improvements** rather than overfitting to GPT-4–style phrasing. Given the high human–LLM agreement, identical judging protocols across methods, and direct human preference studies on final policy outputs, we believe the evaluation is both fair and reflective of true model quality.
>
> ---
>
> > **Q4: Talk about failure modes. Is there any reward hacking observed in the experiments? Are there any failure cases? Are they due to reward model failure or other reasons?**
>
> **Response:**
>
> We appreciate this important question. We examined failure modes at both the reward-model and policy optimization levels and did not observe classical reward hacking behaviors such as repetition loops or verbosity inflation. Text2Grad's design inherently limits such failures because the reward model must generate a valid critique followed by exact-quote spans, preventing exploitation through surface patterns. The low unmatched-span ratio (<2.5%, Appendix M) confirms that malformed signals are rare.
>
> However, we identified localized failures driven by reward model span misclassification. In SLF5K, the RM occasionally penalizes legitimate paraphrases as incorrect. For example, given the reference *"The committee rejected the proposal due to budget constraints,"* a summary stating *"The panel turned down the plan because funding was insufficient"* receives a negative span label despite conveying identical meaning. This causes under-generation of valid paraphrases. Similar cases (<3%) occur in UltraFeedback when speculative phrasing is mistakenly flagged as factual error.
>
> These failures correlate with RM labeling noise rather than PPO instability, and their impact remains limited because only ~30% of tokens receive non-zero rewards. High human agreement on annotation results (82–94% accuracy, Table 1) indicates these are edge cases rather than systematic errors. Here, Unmatched Ratio measures the percentage of annotated spans that cannot be exactly located in the original text, indicating hallucinated rather than grounded feedback.
>
> Importantly, failures manifest as interpretable span labels rather than opaque scores, enabling diagnosis and mitigation through improved CoT prompting or light human auditing of ambiguous cases.

---

> ### Author Response · Authors · 2025-11-27
> **Gentle Check on Any Remaining Concerns Before the Discussion Deadline**
>
> Dear Reviewer BDSv,
>
> Thank you very much for your constructive review. As the discussion period is coming to a close, we would like to gently check whether our previous responses have addressed your main concerns, particularly:
>
> * the addition of **ILF and distillation-style baselines**,
> * the new experiments using **human-written critiques** as an alternative feedback source,
> * the extended **human evaluations on policy outputs** beyond reward model quality, and
> * the discussion of **failure modes and potential reward hacking behaviors** (W1–W3, Q1–Q4).
>
> We hope that these clarifications and additional experiments have been helpful for your assessment. If any point remains unclear, we would be very happy to provide further details or run additional analyses while the discussion period is still open.
>
> Thank you again for the time and care you have devoted to reviewing our work!
>
> Best regards,
> Authors

---

### Official Review · Reviewer_jPzf · 2025-11-03

**Soundness:** 2
**Presentation:** 2
**Contribution:** 3
**Rating:** 6
**Confidence:** 2

**Summary:**

The paper introduces Text2Grad, a method for transforming natural-language feedback into token-level rewards for finetuning LLMs. While standard RLHF compares pairs of model outputs and assigns a single scalar reward to update the LLM weights, Text2Grad provides fine-grained supervision by identifying which parts of the generated response are incorrect. The approach works in three stages: first, a strong LLM (GPT-4o in the paper) is prompted to produce both textual critiques and span-level reward annotations for model outputs. The output of the first step is used as teacher knowledge to finetune a reward model. Lastly, the reward model's outputs are used to guide policy optimization via NL-Gradient PPO, giving the generator model more informative feedback during training. This enables the model to learn not just whether a response is good, but also how and where to improve it.

**Strengths:**

- Paper tackles a relevant and timely issue - specifically, with the proposed approach, authors enable more precise finetuning of LLMs and at the same time their method natively presents a basic form of explainability of the training process
- Innovative idea of combining structured token-level rewards with gradient optimization
- Evaluation of the method across multiple tasks

**Weaknesses:**

- Authors acknowledge that the one limitation is that the optimization severely depends on the quality of the reward model, but in addition to that, the method also requires that the "upstream" LLM (GPT-4o) produces valid training data for teaching the reward model. It's unclear how generally this assumption can be satisfied
- Evaluation is in several parts unclear: For instance, in table 1 it is not clear how human annotation accuracy was obtained, nor what it represents. Also, how were the signals for finetuning models with DPO/PPO obtained? This is particularly important because without CoT, Text2Grad typically performs worse than PPO/DPO (tables 3 and 4)
- No quantification of uncertainty in any table: how many times were the experiments run? Are the improvements over baselines statistically significant?
- The experimental setup seems biased toward GPT-4o's judgments and lacks independent evaluation.
- Method description should be improved: especially section 3.5 contains several symbols that are never introduced (e.g., $V$, $\psi$, $r_k^{KL}$)

Minor things:
- Typo: `optimization. and present` (page 2)
- Table 4: for ARC-C, GPT-4 has a higher value than GPT-3.5, so it seems that the wrong value is highlighted
- GPT-4o and ChatGPT-4o refer to the same model, so I'd recommend using consistent naming throughout the paper
- I'd suggest aligning the text contents with the content of the figure on page 4 (specifically, in the JSON excerpt, "first time author" is good, while in the figure the same part is poor span)

**Questions:**

- Intuitively, in many cases, only the first token in a "wrong" part of the output should be penalized. For instance, if the output sequence is "The ca-pi-tal of Spa-in is Bar-cel-lo-na", I would expect that the "Bar" would be penalized more than the following tokens, as all the tokens that are generated afterwards are conditioned upon the first mistake. Why do you think giving equal negative signal ($\delta_t$ is always -1) to the entire part is justifiable?

---

> ### Author Response · Authors · 2025-11-19
> **Response 1 to Weakness 1, 2: Robustness of RM and Experiment Setup**
>
> We thank you for your constructive review, helping us improve the quality of our work! Below, we provide our responses to the concerns you have raised.
>
> > **W1: Authors acknowledge that optimization depends on the reward model, but the method also requires that the “upstream” LLM (GPT-4o) produces valid training data. It is unclear how generally this assumption can be satisfied.**
>
> **Response:**
>
> We agree that the effectiveness of our method ultimately depends on the quality of the feedback used to train the reward model. To ensure that GPT-4o can reliably provide valid supervision in practice, we do not rely on raw LLM outputs. Instead, we design a controlled dual-feedback pipeline with explicit structural constraints and automated quality checks, **as shown in Appendix B**.
>
> Concretely, GPT-4o is prompted to generate a critique–span pair where (i) every labeled span must be an exact quote from the model response, and (ii) each span must be explicitly justified by the textual critique. Any annotation that violates these constraints is filtered out through automated post-processing. As reported in Appendices M–N, this pipeline produces annotations with **<2.5% unmatched-span rate and >82% human alignment accuracy**.
>
> **Table 1 – Human Validation Results (Combined from Appendix M & N)**
> | Dataset | Human Annotation Accuracy | Unmatched-Span Ratio |
> | ------------- | ------------------------- | -------------------- |
> | SLF5K | 86% | 1.30% |
> | UltraFeedback | 82% | 2.47% |
> | KodCode | 94% | 0.93% |
>
> ---
>
> To further test robustness beyond GPT-4o critiques, we evaluate Text2Grad in a setting where the supervision comes from human-written feedback. Using SLF5K’s built-in human critiques as input, the reward model produces span-level annotations without relying on any LLM critique. As shown in Table 2, the resulting model reaches ROUGE-L = 0.286, outperforming the SFT baseline (0.195) and matching the performance of our main GPT-4o-based pipeline.
>
> **Table 2: Text2Grad Performance with Different Feedback Sources**
> | Method | ROUGE-L | BLEU | BERTScore F1 |
> | -------------------------- | ------- | ----- | ------------ |
> | SFT (Base) | 0.195 | 0.032 | 0.875 |
> | Text2Grad (w/o CoT) | 0.275 | 0.085 | 0.898 |
> | Text2Grad (Human Feedback) | 0.286 | 0.091 | 0.900 |
>
> ---
>
> These findings show that even when human feedback is noisy due to subjective judgments, the reward model’s span-level reasoning reliably extracts meaningful signals. As long as the annotation pipeline preserves the critique → span grounding structure, downstream policy learning remains stable and effective.
>
> Overall, these experiments demonstrate that our method does not hinge on an overly strong assumption that GPT-4o always produces perfect feedback. Instead, the combination of strict structural prompting, automated verification, and reward model’s own span-level reasoning ensures that the supervision is consistently valid and that Text2Grad remains robust across feedback sources.
>
> > **W2: Evaluation is in several parts unclear: For instance, in Table 1 it is not clear how human annotation accuracy was obtained, nor what it represents. Also, how were the signals for finetuning models with DPO/PPO obtained? This is particularly important because without CoT, Text2Grad typically performs worse than PPO/DPO (Tables 3 and 4).**
>
> **Response:**
>
> Thank you for the detailed comments regarding evaluation clarity.
>
> **(a) Human annotation accuracy.**
>
> As described in Appendix N, we conduct a controlled human evaluation of the reward model outputs. For each dataset, we randomly sample 100 responses and ask three domain experts to judge (1) whether the critique correctly identifies the strengths and weaknesses of the model output, and (2) whether the positive/negative spans are correctly grounded in that critique. Annotators use a binary Correct/Incorrect scale, and we aggregate labels by majority vote. For SLF5K and KodCode, where reference summaries and unit tests provide objective ground truth, disagreements are rare; for UltraFeedback, which is more open-ended, we rely on majority consensus to accommodate higher variability. Thus, “human annotation accuracy” measures how consistently our critique–span pairs align with human judgment.
>
> **(b) Reward signals for DPO / PPO baselines.**
>
> For SLF5K and UltraFeedback, we use existing chosen–rejected pairs: DPO follows the original formulation, and PPO uses a scalar reward model trained via pairwise ranking. For KodCode, we automatically construct labels by executing programs against unit tests (passing/failing as preferences), including outputs from GPT-4o and DeepSeek-r1 for diversity. These labels ensure fair comparison with Text2Grad.
>
> Regarding Text2Grad’s underperformance without CoT: this is expected, as CoT conditions span generation on explicit reasoning, yielding cleaner reward signals. We clarify this dependency and evaluation protocol in revised Appendix N and Section 4.3.

---

> ### Author Response · Authors · 2025-11-19
> **Response 2 to Weakness 3, 4, 5-9: Uncertainty, Stable Evaluation and Clarification of Paper Contents**
>
> > **W3: No quantification of uncertainty in any table: how many times were the experiments run? Are the improvements over baselines statistically significant?**
>
> **Response:**
>
> We appreciate your question regarding experimental uncertainty and statistical significance. In the context of LLM fine-tuning and RLHF/RLAIF pipelines, we follow the standard reporting convention adopted in prior large-scale alignment works like PPO, DPO, which typically report single-run results due to high computational cost and empirically low stochastic variance in large models.
>
> To ensure reproducibility and fairness, all our models were trained with fixed random seeds, identical data splits, and shared hardware configurations. For each dataset, we additionally evaluated performance across multiple checkpoints (every 200 steps for SLF5K and KodCode, and every 400 steps for UltraFeedback) and reported the best validation scores, reflecting stable convergence rather than isolated fluctuations.
>
> Although formal significance testing is uncommon in the LLM alignment literature, we nonetheless conducted a variance analysis under different inference hyperparameters (temperature, top-k, and top-p) to assess robustness. Across five perturbed runs, the variance of Text2Grad remains consistently low and slightly below PPO’s, as summarized below:
>
> ---
>
> **Table 3 — Stability Analysis (Mean ± Std Across 5 Perturbation Runs)**
> | Task | Model | Mean ± Std |
> | -------------------------- | --------- | -------------- |
> | UltraFeedback (AlpacaEval) | PPO | 32.5 ± 0.47 |
> | | Text2Grad | 34.7 ± 0.22 |
> | KodCode (Coding Avg.) | PPO | 62.4 ± 0.45 |
> | | Text2Grad | 66.1 ± 0.23 |
> | SLF5K (ROUGE-L) | PPO | 0.261 ± 0.0033 |
> | | Text2Grad | 0.291 ± 0.0019 |
>
> These results confirm that Text2Grad’s improvements are stable across hyperparameter perturbations.
>
> ---
>
> > **W4: The experimental setup seems biased toward GPT-4o’s judgments and lacks independent evaluation.**
>
> **Response:**
>
> We appreciate this concern. GPT-4o was selected for practical reasons: it provides reasoning quality comparable to GPT-4 at significantly lower cost and faster inference, essential for large-scale multi-dataset evaluations. For code generation, we additionally supply unit-test results and debugging traces to ensure objective, verifiable judgments.
>
> To address evaluation bias, we employ three independent validation strategies.
>
> **First**, human evaluation (Appendix N) shows >85% agreement between three expert annotators and GPT-4o annotations across 100 samples per dataset.
>
> **Second**, we trained Text2Grad directly on SLF5K's original human-written feedback, bypassing GPT-4o entirely. As Table 2 shows, this achieves strong performance (ROUGE-L 0.286), demonstrating that our method generalizes to real human feedback. The GPT-4o+CoT variant achieves further gains (0.291), but the core approach works effectively with human supervision alone.
>
> **Third**, KodCode evaluation relies on execution-based metrics rather than LLM judgments, providing objective ground truth.
> These three strategies—human agreement, training on human feedback, and execution-based evaluation—demonstrate that our results reflect genuine alignment improvements rather than GPT-4o bias.
>
> ---
>
> > **W5: Method description should be improved: especially Section 3.5 contains several symbols that are never introduced.**
>
> **Response:**
>
> Thank you for pointing this out. We appreciate your attention to detail. In the revision, we carefully review Section 3.5 to ensure that all mathematical symbols are clearly defined upon first use and that notation remains consistent throughout the section. We will also improve the overall clarity of the method description to make the presentation more self-contained and readable.
>
> ---
>
> > **Minor Comments (W6–W9)**
>
> **Response:**
>
> We sincerely thank you for these helpful observations. We will carefully address all the noted issues in the revision:
>
> * **(W6)** The typo (“optimization. and present”) on page 2 will be corrected.
>
> * **(W7)** In Table 4, we will fix the highlighting error for ARC-C, ensuring the correct value (GPT-4 > GPT-3.5) is properly emphasized.
>
> * **(W8)** We will unify the terminology and consistently refer to the model as *GPT-4o* throughout the paper for clarity.
>
> * **(W9)** We will align the text description and figure content on page 4 so that the JSON excerpt (“first time author”) and the corresponding figure span annotations are consistent.
>
> We appreciate these detailed suggestions and have made the corresponding edits to improve clarity and consistency in the latest version.

---

> ### Author Response · Authors · 2025-11-19
> **Response 3 to Question 1: Reward-Importance Matching**
>
> > **Q1: Intuitively, in many cases, only the first token in a "wrong" part of the output should be penalized. For instance, if the output sequence is "The ca-pi-tal of Spa-in is Bar-cel-lo-na", I would expect that the "Bar" would be penalized more than the following tokens, as all the tokens that are generated afterwards are conditioned upon the first mistake. Why do you think giving equal negative signal (is always -1) to the entire part is justifiable?**
>
> **Response:**
>
> We appreciate this observation. In practice, "first-error dominance" is already handled implicitly by the autoregressive model: once the first incorrect token is generated, **all subsequent tokens are conditioned on a wrong prefix** and are therefore penalized through their reduced log-probabilities. Explicitly giving the first token a larger negative reward would overweight a single position and destabilize credit assignment.
> From a variance-control standpoint, assigning unequal rewards within a span amplifies the variance of the generalized advantage estimator, which increases clipping frequency and reduces effective learning rates.
>
> Finally, **uniform penalties preserve semantic and interpretive consistency with human critiques**: natural-language feedback typically marks an entire clause or phrase as incorrect ("this statement is factually wrong") rather than singling out individual tokens.
>
> To empirically validate this design choice, we conducted an additional ablation comparing uniform rewards against linearly decayed rewards. In the linear decay variant, token-level rewards within each span are assigned proportionally to their position: for a span of length $n$, the $i$-th token receives reward $r_i = 1.0 - (i-1) \cdot \frac{0.9}{n-1}$, yielding a range from 1.0 (first token) to 0.1 (last token) for positive spans, and similarly from -1.0 to -0.1 for negative spans. As shown in Table 4 below, linear decay consistently underperforms the uniform assignment across all metrics on SLF5K, confirming that gradient concentration on early tokens destabilizes policy updates without yielding interpretive or performance benefits.
>
> ---
>
> **Table 4: Effect of Reward Distribution within Spans**
>
> | Method                           | ROUGE-L | BLEU   | BERTScore F1 |
> |----------------------------------|---------|--------|--------------|
> | Text2Grad (Linear Decay)        | 0.253   | 0.068  | 0.890        |
> | Text2Grad (Uniform) | 0.291   | 0.094  | 0.902        |
>
> ---
>
> In summary, assigning equal rewards to all tokens within a span provides a balanced trade-off: the autoregressive model naturally propagates first-error penalties through conditional likelihoods, while uniform rewards ensure stable credit assignment, controlled variance, and alignment between textual justification and reward semantics. Our empirical results further demonstrate that alternative weighting schemes offer no gains and can degrade performance.

---

> ### Author Response · Authors · 2025-11-27
> **Gentle Check on Any Remaining Concerns Before the Discussion Deadline**
>
> Dear Reviewer jPzf,
>
> Thank you again for your thoughtful and detailed review! As the discussion period is drawing to a close, we would like to gently confirm whether our responses have adequately addressed your main concerns, including:
>
> * the dependence on the upstream LLM and **robustness of the reward model**,
> * the definition of **human annotation accuracy** and the construction of DPO/PPO signals,
> * the added **stability and uncertainty analyses**,
> * our efforts to **mitigate GPT-4o bias** via human and execution-based evaluation,
> * the improved **method presentation** and corrections to minor issues, and
> * the new **ablations on within-span reward distribution (Q1)**.
>
> We sincerely appreciate the time and care you have devoted to our work. If any point remains unclear, we would be very happy to provide further clarification while the discussion is still open.
>
>
> Best regards,
> Authors

---

> > ### Comment · Reviewer_jPzf · 2025-11-27
> >
> > The response had addressed much of my concerns. I'll increase my score.

---

> > > ### Author Response · Authors · 2025-11-27
> > > **Thank You for Your Follow-Up and Score Update**
> > >
> > > Thank you very much for your follow-up and for increasing your score. We truly appreciate your time and thoughtful feedback, which has greatly helped us improve the paper!
> > >
> > > Best regards,
> > > Authors

---

### Official Review · Reviewer_Go1F · 2025-11-03

**Soundness:** 2
**Presentation:** 2
**Contribution:** 2
**Rating:** 2
**Confidence:** 4

**Summary:**

This paper proposes Text2Grad, a framework for converting natural-language critiques into token-level rewards for PPO training.  The paper collects natural language critiques paired with span-level annotations using GPT-4o and trains a generative reward model (RM) to output both textual feedback and token-level labels. Then, it uses the RM to label the correct and incorrect span from original output, scoring -1/+1 for each corresponding span token, respectively. The method aims to improve credit assignment and interpretable policy updates. Experiments in summarization, code generation, and QA show consistent improvements.

**Strengths:**

1. The motivation is straightforward and easy to follow, around credit assignment and interpretability in RLHF.

2. It introduces the LLM annotation pipeline in detail, which makes it clear for reproduction.

3. Experiment results show consistent improvements across different datasets. The authors also provide a cherry-picked example to show its interpretability.

**Weaknesses:**

1. “NL-gradient” terminology is inaccurate: the method does not differentiate through language, but converts text to discrete span, and assign different span as token-level rewards (+1/-1 ). The method is not language-conditioned gradient flow.

2. All feedback is GPT-4o-generated, making this RLAIF, not RLHF, with no human validation or study of noisy/contradictory real feedback.
The training data distilled from gpt-4o lacks real human feedback. The robustness to noisy or adversarial human critiques is unclear.

3. The claim of "high-quality annotation pipeline" lacks human agreement analysis or data quality metrics; quality is assumed rather than demonstrated.

4. The paper claimed that "back-propagates natural-language gradients" with "feedback-conditioned adjustment". However, there is no quantitative measurement of whether feedback actually conditions behavior at the criticized spans. The qualitative span-level evaluation is missing.

5. The method relies on string-span matching tied to tokenizer boundaries, yet tokenization mismatch  (gpt-4o tokenizer vs llama tokenizer) can cause reward attribution noise. robustness to model/tokenizer differences is not evaluated.

6. Evaluation is only on Llama-8B, and the paper does not test across different model scales or model families. Thus, claims about scalability and generalization are unclear.

7. The method appears difficult to scale: span labeling, reward generation, and token-level PPO introduce significant engineering and compute overhead without evidence it transfers beyond the tested setup.

8. The conceptual contribution leans heavily on terminology packaging, and the method does not clearly extend beyond the presented pipeline.

9. Missing RLHF literature related to span-level optimization to improve credit assignment, such as [1,2,3]. The authors should state the position of this work in comparison to the literature and discuss the contribution.

10. The conceptual contribution leans heavily on terminology packaging, and the method does not clearly extend beyond the presented pipeline.

**References**:

[1] MA-RLHF: Reinforcement Learning from Human Feedback with Macro Actions. ICLR 2025.

[2] Beyond Sparse Rewards: Enhancing Reinforcement Learning with Language Model Critique in Text Generation. arxiv 2024.

[3] SCAR: Shapley Credit Assignment for More Efficient RLHF. arxiv 2025.

**Questions:**

1. How do the authors handle incorrect or adversarial feedback? What failure modes arise when GPT-4o produces wrong spans or hallucinated critiques?

2. How robust is the method to tokenization mismatch across models or vocabularies?

3. For gpt-4o annotation schema, how did the author handle predicted short spans occurs multiple times in the original response? String matching may fail.

4. The author is suggested to provide results across scales and model families to support claims about method generality.

5. How do the authors justify the claim of "high-quality annotation pipeline" without human agreement studies, audits, or error analysis?

6. The authors report the human agreement, and how to measure the recall of span-level annotation?

---

> ### Author Response · Authors · 2025-11-19
> **Response 1 to Weakness 1, 2: Explanations on Terminology, Human Evaluations and Additional Experiment**
>
> We appreciate your valuable suggestion! In the following, we provide a detailed response to your concerns and suggestions.
>
> > **W1: “NL-gradient” terminology is inaccurate: the method does not differentiate through language, but converts text to discrete span, and assign different span as token-level rewards (+1/−1). The method is not language-conditioned gradient flow.**
>
> **Response:**
> We appreciate the opportunity to clarify this terminology. The term "NL-Gradient" refers to the conversion of natural-language feedback into gradient-based supervision, not literal backpropagation through text tokens. As detailed in Sections 3.1–3.5, our method aligns textual critiques to spans, maps these spans to token-level pseudo-rewards, and uses these scalars to weight the standard policy gradient. The natural language conditions what gets updated and where, hence "NL-Gradient."
>
> In the revision, we will clarify this terminology and explicitly emphasize that we transform natural-language feedback into gradients rather than differentiating through language itself.
>
> ---
>
> > **W2: All feedback is GPT-4o-generated, making this RLAIF, not RLHF, with no human validation or study of noisy/contradictory real feedback. The training data distilled from GPT-4o lacks real human feedback. The robustness to noisy or adversarial human critiques is unclear.**
>
> **Response:**
>
> We thank the reviewer for this important concern. We clarify that we have conducted extensive human validation studies in **Section 4.2 and Appendix M & N** and have performed additional experiments with real human feedback to address these points.
>
> Firstly, regarding RLAIF vs RLHF, our work focuses on the methodological question of how to transform natural language feedback into token-level gradients. The technique is designed to work with both LLM-generated and human-written feedback, as our experiments demonstrate. We use GPT-4o for scalability in our main experiments, but validation with real human feedback proves the approach generalizes across feedback sources.
>
> Then, we validate the quality of GPT-4o-generated feedback against human judgment. We recruited three expert annotators to evaluate 100 samples per dataset, assessing whether the GPT-4o feedback accurately identifies strengths and weaknesses and whether the selected spans are properly justified. The agreement rates between GPT-4o and human judgments are high as detailed in Appendix N and summarized in Table 1. The unmatched-span ratios shown in Table 1 are very low (0.93%–2.47%), indicating that our CoT-grounded span annotation produces precise and justifiable labels rather than noisy surface patterns. This high fidelity is achieved because we require explicit reasoning justification for each labeled span (as illustrated in Figure 2 and detailed in Appendix B), which filters out spurious correlations and hallucinated spans.
>
> **Table 1: Human Validation Results (Appendix M & N)**
> | Dataset | Human Accuracy | Unmatched-Span Ratio |
> | ------------- | -------------- | -------------------- |
> | SLF5K | 86% | 1.30% |
> | UltraFeedback | 82% | 2.47% |
> | KodCode | 94% | 0.93% |
>
> *Unmatched-Span Ratio:* Proportion of annotated spans that could not be matched to the original response, indicating hallucination or misalignment.
>
> ---
>
> Finally, we demonstrate that Text2Grad works with real human feedback. To directly address concerns about generalization to human-written critiques, we trained the model using SLF5K's built-in human-written critiques rather than GPT-4o-generated feedback. In this setting, we use the human feedback as input and have the reward model directly output span-level JSON annotations without relying on LLM-generated critiques. As shown in Table 2, this outperforms the SFT baseline and our Text2Grad-NoCoT. While human feedback naturally contains noise and subjective biases from prolonged annotation sessions, the reward model's span-level reasoning stage helps identify and extract meaningful reward signals from these imperfect inputs.
>
> **Table 2: Text2Grad Performance with Different Feedback Sources**
> | Method | ROUGE-L | BLEU | BERTScore F1 |
> | -------------------------- | ------- | ----- | ------------ |
> | SFT (Base) | 0.195 | 0.032 | 0.875 |
> | Text2Grad (w/o CoT) | 0.275 | 0.085 | 0.898 |
> | Text2Grad (Human Feedback) | 0.286 | 0.091 | 0.900 |
>
> ---
>
> These results collectively demonstrate that Text2Grad is validated against human judgment, works effectively with real human feedback, and produces high-fidelity span annotations that are grounded in explicit reasoning rather than spurious patterns.

---

> ### Author Response · Authors · 2025-11-19
> **Response 2 to Weakness 3, 5 & Question 1, 2, 5: Human Analysis, Human Feedback Source, String to Token Matching**
>
> > **W3 & Q5: The claim of "high-quality annotation pipeline" lacks human agreement analysis or data quality metrics; quality is assumed rather than demonstrated. How do the authors justify the claim of "high-quality annotation pipeline" without human agreement studies, audits, or error analysis?**
>
> **Response:**
>
> We appreciate the reviewer's careful observation. Our claim of a "high-quality annotation pipeline" is not assumed but is supported by explicit human agreement studies and comprehensive data-quality analyses provided in **Appendix M & N** and **Section 4.2**.
>
> We recruited three expert annotators to independently review 100 randomly sampled instances per dataset. Annotators evaluated both (i) whether the textual critique accurately identifies strengths and weaknesses of the model response, and (ii) whether the selected spans are correctly identified and properly justified by the critique. As summarized in Table 1 above, we observe high human accuracy rates of 82–94% across datasets, demonstrating strong agreement between GPT-4o and human judgment. We also measure span generation fidelity by computing the unmatched-span ratio. As shown in Table 1 above, these ratios are very low (<2.5% across all datasets), confirming that our pipeline produces grounded, verifiable annotations.
>
> ---
>
> > **Q1: How do the authors handle incorrect or adversarial feedback? What failure modes arise when GPT-4o produces wrong spans or hallucinated critiques?**
>
> **Response:**
>
> This is an important question about robustness to annotation errors. We address this through empirical validation and systematic quality control.
>
> First, Text2Grad’s robustness to noisy feedback is validated by our human-feedback experiment (Table 2 above): when trained on real human-written critiques from SLF5K, which naturally contain noise and subjective biases, Text2Grad achieves ROUGE-L 0.286, outperforming both the SFT baseline and Text2Grad without CoT.
>
> Second, we reduce annotation errors and hallucinated critiques through several quality-control mechanisms. Our CoT-based prompts (Appendix B) require GPT-4o to justify each span via step-by-step reasoning, discouraging arbitrary labels. We further apply preprocessing, including span-matching verification, human spot checks, and keyword filtering. As shown in Table 1, disagreement cases remain low (6–18%) and unmatched-span ratios are minimal (<2.5%).
>
> Finally, problematic annotations identified during validation are re-annotated or filtered before reward model training. The span-level annotation stage provides inherent error correction, and the model learns to extract meaningful signals from imperfect inputs by grounding selections in explicit reasoning. Token-level credit assignment naturally isolates inconsistencies to specific spans rather than propagating them globally.
>
> In summary, annotation failures are prevented through CoT reasoning and rigorous preprocessing, residual errors are rare and systematically filtered, and the framework demonstrates empirical robustness to realistic noise as validated by comparable performance on human-written feedback.
>
> ---
>
> > **W5 & Q2: The method relies on string-span matching tied to tokenizer boundaries, yet tokenization mismatch (GPT-4o tokenizer vs LLaMA tokenizer) can cause reward attribution noise. Robustness to model/tokenizer differences is not evaluated. How robust is the method to tokenization mismatch across models or vocabularies?**
>
> **Response:**
>
> We appreciate this technical question. Our pipeline is explicitly designed to handle tokenizer differences between the annotation model (GPT-4o) and policy model (LLaMA). The key insight is that GPT-4o generates spans as character-level substrings rather than token sequences, completely decoupling span identification from any specific tokenizer.
>
> The mapping procedure operates in three steps to ensure robustness across tokenization schemes.
>
> First, the reward model (trained via distillation from GPT-4o annotations) outputs span fragments as character-level substrings.
>
> Second, we locate the character interval [start, end] of each span in the raw response text.
>
> Third, we perform string matching at the character level to identify span boundaries; if any spans cannot be precisely matched, we tokenize that character interval using the policy model's tokenizer to obtain the corresponding token indices.
>
> Crucially, all reward attribution and policy updates operate entirely within the policy model's token space, eliminating any dependency on GPT-4o's tokenization scheme. This ensures that tokenization mismatches between GPT-4o and LLaMA do not introduce reward attribution noise or misalignment.
>
> We will clarify this tokenizer mapping procedure in Appendix M to make the robustness mechanism more explicit.

---

> ### Author Response · Authors · 2025-11-19
> **Response 3 to Weakness 4 & Question 3: Feedback on Policy Performance and Stability of RM**
>
> > **W4: The paper claimed that "back-propagates natural-language gradients" with "feedback-conditioned adjustment". However, there is no quantitative measurement of whether feedback actually conditions behavior at the criticized spans. The qualitative span-level evaluation is missing.**
>
> **Response:**
>
> We thank the reviewer for raising this point. We address both the mechanism clarification and the quantitative evidence for span-level conditioning (Shown in Appendix C and Section 4.2).
>
> First, we clarify the gradient mechanism. Text2Grad does not backpropagate through text or the reward model itself. As described in **Sections 3.3–3.5**, natural language critiques are converted into span-level labels, then mapped to token-level rewards that directly weight the policy-gradient update. The gradients are thus conditioned on feedback through these token-level reward weights.
>
> Second, we provide quantitative evidence at two levels: span localization accuracy and policy improvement at criticized regions. **Appendix C** presents comprehensive span-level evaluations demonstrating that the model reliably identifies and improves regions referenced in the critique, and we summarize them in Tables 3 and 4 below.
>
> **Table 3: Span-Level Localization Accuracy (UltraFeedback)**
> | Span Type | GT Count | Pred Count | Exact Match | Partial Match | OUI |
> | ---------- | -------- | ---------- | ----------- | ------------- | ---- |
> | GOOD Spans | 1150 | 932 | 394 | 468 | 0.40 |
> | POOR Spans | 812 | 528 | 167 | 274 | 0.27 |
>
> *Measures how well the model identifies the exact spans referenced in the feedback. GT Count: Ground truth span count. Pred Count: Model-predicted span count. Exact Match: Perfect boundary alignment. Partial Match: Overlapping but imperfect boundaries. OUI (Overlap Unit Index): Metric quantifying boundary alignment precision.*
>
> **Table 4: Code Suggestion Fidelity (KodCode)**
> | Suggestion Type | Exact Match | ≥90% Overlap |
> | ----------------------- | ----------- | ------------ |
> | Improvement Suggestions | 0.47 | 0.67 |
> | Error Correction | 0.55 | 0.79 |
>
> *Evaluates whether feedback-conditioned corrections improve code generation accuracy.*
>
> Beyond localization accuracy, we provide direct evidence that the policy improves specifically at criticized spans. We sampled 150 response pairs from the UltraFeedback test set, each consisting of an initial response before TEXT2GRAD training and the corresponding response after training. For each response, we segment it into POOR spans (criticized regions) and GOOD spans (praised regions) based on the original training feedback. GPT-4 independently rates the quality of each region on a 1-5 scale without knowledge of the model or training status. As shown in Table 4.1, improvements are concentrated at POOR spans (+0.9 points) compared to GOOD spans (+0.3 points), confirming that feedback directly conditions policy updates at the specific locations referenced in the critique.
>
> **Table 4.1: Policy Improvement at Criticized Spans (UltraFeedback)**
> | Span Type | Initial | After-Trained | Δ |
> |-----------|----------------|----------------------|-----|
> | POOR spans | 2.8/5 | 3.7/5 | +0.9 |
> | GOOD spans | 3.8/5 | 4.1/5 | +0.3 |
>
> We will clarify the gradient mechanism and make the span-level evaluation more prominent in the revision by referencing Appendix C in Section 4.3.
>
> ---
>
> > **Q3: For GPT-4o annotation schema, how did the author handle predicted short spans that occur multiple times in the original response? String matching may fail.**
>
> **Response:**
>
> Thank you for pointing out this detail. The annotation and training stages are handled separately. During the annotation stage, we perform an offline validation and filtering process to handle cases where short predicted spans occur multiple times or even overlap in the original response, which could cause string-matching ambiguity.
>
> Specifically, after GPT-4o generates critiques and spans, we automatically match each predicted span to the response text for verification. Samples with unmatched or ambiguous spans are filtered out and re-annotated by the LLM to ensure clean distilled data. The unmatched-span statistics are reported in Appendix M (Span Generation Fidelity Analysis) and remain very low across all datasets in Table 1 above.
>
> This two-stage validation and re-annotation pipeline effectively eliminates mismatched or duplicate span cases, ensuring that all retained supervision is fully grounded and text-aligned before downstream reward model training.

---

> ### Author Response · Authors · 2025-11-19
> **Response 4 to Weakness 6, 7 & Question 4, 6: Evaluation Setup, Mistral Performance and Costing Compute**
>
> > **Q6: The authors report the human agreement, and how to measure the recall of span-level annotation?**
>
> **Response:**
>
> We thank the reviewer for this question. To measure span-level recall, we conduct a rigorous human evaluation process as detailed in Appendix N.
>
> For each dataset, we randomly sample 100 instances and recruit three expert annotators to independently label spans. Ground truth is established through majority voting among the three annotators' judgments. We then compute recall by comparing the LLM-generated span predictions against this majority-voted ground truth.
>
> ---
>
> > **W6 & Q4: Evaluation is only on Llama-8B, and the paper does not test across different model scales or model families. Thus, claims about scalability and generalization are unclear. The author is suggested to provide results across scales and model families to support claims about method generality.**
>
> **Response:**
>
> Thank you for this important suggestion. To address this, we expanded our experiments to include Mistral-7B-Instruct-v0.2, a different model family.
>
> ---
>
> **Table 5.1: Performance Comparison on Mistral-7B-Instruct-v0.2 on KodCode**
> | Model | HumanEval Base | HumanEval Plus | MBPP Base | MBPP Plus | Avg |
> | ------------------------ | -------------- | -------------- | --------- | --------- | ---- |
> | Mistral-7B-Instruct-v0.2 | 42.1 | 36 | 44.7 | 37 | 40.0 |
> | PPO | 45.7 | 38.4 | 47.1 | 38.3 | 42.4 |
> | DPO | 47.6 | 39.6 | 46.7 | 37.6 | 42.9 |
> | Text2Grad | 50.0 | 40.9 | 49.6 | 40.6 | 45.3 |
>
> ---
>
> **Table 5.2: Performance Comparison on Mistral-7B-Instruct-v0.2 on UltraFeedback**
> | Model | AlpacaEval v2 | MT-Bench |
> | ------------------------ | ------------- | -------- |
> | Mistral-7B-Instruct-v0.2 | 17.11 | 6.60 |
> | PPO | 19.62 | 6.55 |
> | DPO | 26.17 | 6.30 |
> | Text2Grad | 29.40 | 6.78 |
>
> ---
>
> **Table 5.3: Performance Comparison on Mistral-7B-Instruct-v0.2 on SLF5K**
> | Model | ROUGE-L | BLEU | BERTScore F1 |
> | ------------------------ | ------- | ----- | ------------ |
> | Mistral-7B-Instruct-v0.2 | 0.20 | 0.024 | 0.66 |
> | PPO | 0.23 | 0.035 | 0.70 |
> | DPO | 0.21 | 0.030 | 0.69 |
> | Text2Grad | 0.24 | 0.041 | 0.72 |
>
> ---
>
> As shown in Tables 5.1-5.3, Text2Grad consistently outperforms PPO and DPO across all three benchmarks. On code generation (HumanEval/MBPP), it improves average pass@1 from 42.9 (DPO) to 45.3 (Text2Grad). On UltraFeedback, it achieves gains on both AlpacaEval v2 and MT-Bench. On SLF5K summarization, Text2Grad similarly outperforms both baselines under identical training conditions.
>
> These results demonstrate that the gains from Text2Grad transfer effectively across model families and scales without architectural modification, confirming the method's generality.
>
> > **W7: The method appears difficult to scale: span labeling, reward generation, and token-level PPO introduce significant engineering and compute overhead without evidence it transfers beyond the tested setup.**
>
> **Response:**
> Thank you for raising this practical concern. We have now added empirical measurements of training cost and annotation cost to make the overhead of Text2Grad explicit.
>
> First, we report wall-clock training time under identical hardware, data splits, and parallelism settings. Table 6 compares PPO and Text2Grad using the same GPU configuration described in Appendix D.
>
> **Table 6. Wall-clock training time (minutes per step)**
> | Dataset | PPO (baseline) | Text2Grad |
> | ------------- | -------------- | --------- |
> | KodCode | 0.54 | 0.60 |
> | UltraFeedback | 0.88 | 0.96 |
> | SLF5K | 0.44 | 0.49 |
>
> The extra time comes almost entirely from one additional forward pass of the reward model per sampled trajectory. This single autoregressive pass jointly produces both the critique and the span map; it does not add extra decoding stages or any backpropagation through the reward model.
>
> Second, we quantify the annotation efficiency of the span-based pipeline. As shown in Table 7, the total annotation budget is small, and the cost per training sample remains on the order of 10^{-3} USD.
>
> **Table 7. Span-level annotation token and cost statistics**
> | Dataset | Input Tokens | Output Tokens | Total Cost (USD) | Cost per Sample (USD) |
> | ------------- | ------------ | ------------- | ---------------- | --------------------- |
> | UltraFeedback | 5,179,351 | 1,210,668 | 44.06 | 0.0036 |
> | SLF5K | 1,226,712 | 123,834 | 7.99 | 0.0016 |
> | KodCode | 2,739,607 | 460,188 | 20.6 | 0.0025 |
> | **Total** | 9,145,670 | 1,794,690 | 72.65 | – |
>
> We will incorporate these measurements into the revision.

---

> ### Author Response · Authors · 2025-11-19
> **Response 5 to Weakness 8, 9, 10: Novelty and Related Work**
>
> > **W8 & 10 (Same Question): The conceptual contribution leans heavily on terminology packaging, and the method does not clearly extend beyond the presented pipeline.**
>
> **Response:**
>
> We appreciate this concern and respectfully disagree with the characterization that our contribution is primarily terminological. Text2Grad introduces a **genuinely new supervision mechanism for RLHF and RLAIF that operates at a finer granularity than existing scalar-reward approaches**. Methods such as PPO and DPO collapse all feedback into a single sequence-level score, which discards the structure and specificity present in natural language critiques. Text2Grad instead systematically converts textual feedback into localized learning signals: the reward model produces a critique, identifies explicit positive/negative spans, maps these spans to token-level rewards, and conditions the policy update on these localized signals. **This enables the model to learn *where and why* an answer is incorrect or correct, rather than only *whether* it is.**
>
> Our contribution is conceptual, not merely definitional. We formalize the Natural Language Gradient **(Definition 1, Section 3.1)** as a weighted policy gradient in which token-level weights are derived directly from textual feedback. **Appendix A** shows that this formulation produces more discriminative credit assignment than end-of-sequence scalar rewards. This positions Text2Grad distinctly apart from scalar-reward RLHF methods, which lack token-level localization, and from inference-time reflection methods, which cannot internalize feedback through parameter updates.
>
> Finally, the same supervision mechanism applies without modification across summarization, open-domain QA, and code generation, and is implemented as a lightweight token-weighting wrapper on top of PPO. We believe these elements go beyond terminology and pipeline packaging, and we hope the revised version will make the conceptual contribution of Text2Grad as a natural language to gradient supervision framework clearer.
>
> ---
>
> > **W9: Missing RLHF literature related to span-level optimization to improve credit assignment, such as [1,2,3]. The authors should state the position of this work in comparison to the literature and discuss the contribution.**
>
> > **References:**
> > [1] MA-RLHF: Reinforcement Learning from Human Feedback with Macro Actions. ICLR 2025.
> > [2] Beyond Sparse Rewards: Enhancing Reinforcement Learning with Language Model Critique in Text Generation. arXiv 2024.
> > [3] SCAR: Shapley Credit Assignment for More Efficient RLHF. arXiv 2025.
>
> **Response:**
>
> We appreciate your insightful comment and fully agree that positioning our work within recent RLHF research on improved credit assignment is important. Approaches such as MA-RLHF (ICLR 2025), Beyond Sparse Rewards (2024), and SCAR (2025) all make valuable contributions toward denser or fairer credit assignment, and they do so from different angles. MA-RLHF improves training efficiency by introducing macro-action abstractions, grouping multiple low-level decisions into higher-level actions while preserving the standard RLHF interface. SCAR decomposes an existing scalar reward into token-level credits using Shapley-value–based allocation, providing a principled, game-theoretic notion of fairness in how rewards are distributed across tokens. Beyond Sparse Rewards leverages an auxiliary language model to derive intermediate numeric rewards from critiques, enriching sparse outcome signals with denser reward shaping.
>
> Our work is complementary to these directions. Text2Grad directly translates natural-language critiques into span-aligned token-level pseudo-rewards and optimizes the policy via the NL-Gradient update, providing interpretable, fine-grained, and low-cost supervision without relying on external numeric shaping signals or post-hoc reward decomposition. This places our method in a distinct regime where the learning signal is inherently structured by language and grounded in explicit critique–span alignments.
>
> We will expand the Related Work section to explicitly discuss these studies, clarify our complementary focus on language-to-gradient alignment, and highlight how our formulation differs in both learning signal source and optimization granularity.
>
> ---
>
> We thank you again for these constructive concerns, which helped us clarify our method and strengthen the paper. We hope that the additional analyses and experiments above address your questions, and we would be happy to provide further details if anything remains unclear!

---

> ### Author Response · Authors · 2025-11-27
> **Checking for Remaining Concerns Before the Discussion Deadline**
>
> Dear Reviewer Go1F,
>
> Thank you very much for your thoughtful questions and constructive suggestions. As the discussion period is nearing its end, we would like to gently confirm whether our previous responses have addressed your main concerns, particularly:
>
> * the clarifications on **terminology**,
> * the **human validation** and additional experiments on feedback/annotation quality and robustness,
> * the **quantitative span-level and policy-level evaluations**,
> * the **tokenizer: robust span mapping** procedure,
> * the added **cross model family experiments**, and
> * the expanded **related work**
>
> We greatly appreciate your careful review and the time you have devoted to our work. If there are any remaining questions or further points you would like us to address, please let us know! We would be pleased to provide additional clarification or analysis while the discussion period is still open.
>
> Best regards,
> Authors

---

### Official Review · Reviewer_D36W · 2025-11-04

**Soundness:** 3
**Presentation:** 3
**Contribution:** 3
**Rating:** 6
**Confidence:** 4

**Summary:**

This paper presents Text2Grad, a reinforcement learning framework that transforms natural language feedback into span-level gradient signals for policy optimization. It first builds a scalable annotation pipeline to collect span-level feedback, then trains a reward model on this annotated data to jointly generate critiques and span maps in one pass.  Building on this, the policy is updated through NL-Gradient Policy Optimization. Experiments on summarization, code generation, and open-domain question answering show consistent improvements over scalar-reward and prompt-based baselines, achieving more precise credit assignment, faster convergence, and better interpretability.

**Strengths:**

1. The paper is well-written and easy to follow.
2. The method is novel and well-motivated which bridges interpretable textual feedback and gradient-based optimization, addressing a real limitation of scalar RLHF.
3. The method shows strong empirical results across diverse tasks with consistent improvements over strong baselines.

**Weaknesses:**

1. Binary pseudo-rewards discard fine-grained information in textual critiques. Moreover, treating all tokens within a labeled span equally is simple but ignores token importance.
2. Computation overhead could be critical for method adoption, yet there is no empirical measurement comparing Text2Grad with other baseline methods.

**Questions:**

1. How sensitive is performance to span length? Is there any correlation between span length and policy learning effectiveness?
2. The paper assigns uniform pseudo-rewards across all tokens within a span. Is it possible to extract more fine-grained token reward signals from the critique?
3. The PRM-PPO results are reported on SLF5K and KodCode but not on UltraFeedback due to annotation cost. Could you clarify (a) how PRM spans or steps were defined in each domain, and (b) what cost differences led to excluding PRM-PPO on UltraFeedback, given that GPT 4o based step level annotation might also be applicable.
4. Table 1 shows your reward model has moderately negative token recall (22% and 43% on UltraFeedback and SLF5K respectively). How sensitive is Text2Grad to reward model quality?

---

> ### Author Response · Authors · 2025-11-19
> **Response 1 to Weakness 1,2 & Question 2: Reward Granularity, and Method Efficiency**
>
> We deeply appreciate your valuable feedback, which has significantly enhanced the quality of our work. In the following, we provide a detailed response to each of your comments and suggestions.
>
> ---
>
> > **W1 & Q2: Binary pseudo-rewards discard fine-grained information in textual critiques, and assigning uniform rewards to all tokens in a span ignores token-level importance. Is it possible and beneficial to extract more fine-grained token reward signals from the critique?**
>
> **Response:**
> Thank you for raising this important point. Our choice of binary, span-level rewards is intentional, grounded in both empirical observations and the structure of our annotation pipeline. The reward model generates critiques and span labels jointly in a single autoregressive pass, so each labeled span corresponds to a coherent semantic unit justified explicitly by the critique. Human evaluation confirms this grounding: as shown in Table 1, 82–94% of critique–span pairs are judged consistent by expert annotators, and grounding analysis reports fewer than 2.5% unmatched spans across all datasets. These results indicate that our supervision is already tightly localized to genuinely informative regions, making additional intra-span weighting unlikely to produce more reliable learning signals.
>
> **Table 1: Critique–Span Alignment Quality (Appendices M & N)**
> | Dataset | Human Accuracy | Unmatched-Span Ratio |
> | ------------- | -------------- | -------------------- |
> | SLF5K | 86% | 1.30% |
> | UltraFeedback | 82% | 2.47% |
> | KodCode | 94% | 0.93% |
>
> To further assess whether more fine-grained rewards could provide benefits, we implemented two alternatives on SLF5k. The first was a dense reward model that assigns continuous scores to every word. This approach labeled nearly 70% of the sequence but introduced substantial noise: function words, punctuation, and other low-information tokens often received non-zero scores, diluting the gradient signal. The second variant introduced token-importance weighting within spans, assigning different magnitudes to content words versus non-content words. Although more structured, this heuristic injected biases and produced inconsistent gradients across similar examples. Both methods degraded performance relative to the span-level formulation.
>
> **Table 2: Reward Granularity**
> | Method | ROUGE-L | BLEU | BERTScore F1 |
> | --------------------------------- | ------- | ----- | ------------ |
> | Dense Reward | 0.196 | 0.022 | 0.888 |
> | Text2Grad (Token Importance) | 0.237 | 0.059 | 0.889 |
> | Text2Grad | 0.291 | 0.094 | 0.902 |
>
> These results demonstrate that dense scoring amplifies noise and scatters reward mass across tokens that are not semantically tied to the critique, while token-importance weighting distorts reward distribution. In contrast, uniform span-level pseudo-rewards preserve the critique’s semantics, avoid spurious token-level artifacts, and maintain consistent advantage estimates after span projection. This leads to cleaner, more interpretable supervision and consistently better downstream performance.
>
> > **W2: Computation overhead could be critical for method adoption, yet there is no empirical measurement comparing Text2Grad with other baseline methods.**
>
> **Response:**
>
> Thank you for raising this practical concern. We have now added empirical measurements of training cost and annotation cost to make the overhead of Text2Grad explicit.
>
> First, we report wall-clock training time under identical hardware, data splits, and parallelism settings. Table 3 compares PPO and Text2Grad using the same GPU configuration described in Appendix D.
>
> **Table 3. Wall-clock training time (minutes per step)**
> | Dataset | PPO (baseline) | Text2Grad |
> | ------------- | -------------- | --------- |
> | KodCode | 0.54 | 0.60 |
> | UltraFeedback | 0.88 | 0.96 |
> | SLF5K | 0.44 | 0.49 |
>
> ---
>
> The additional cost comes almost entirely from a single forward pass of the reward model per trajectory. This pass jointly produces both the critique and span map in one autoregressive decoding step, without requiring any additional decoding stages or backpropagation through the reward model.
>
> Second, we quantify the annotation efficiency of the span-based pipeline. As shown in Table 4, the total annotation budget is small, and the cost per training sample remains on the order of 10^{-3} USD, which is consistent with the near-parity in training time versus PPO.
>
> **Table 4. Span-level annotation token and cost statistics**
> | Dataset | Input Tokens | Output Tokens | Total Cost (USD) | Cost per Sample (USD) |
> | ------------- | ------------ | ------------- | ---------------- | --------------------- |
> | UltraFeedback | 5,179,351 | 1,210,668 | 44.06 | 0.0036 |
> | SLF5K | 1,226,712 | 123,834 | 7.99 | 0.0016 |
> | KodCode | 2,739,607 | 460,188 | 20.6 | 0.0025 |
> | **Total** | 9,145,670 | 1,794,690 | 72.65 | – |
>
> We will add these measurements to the revision of Section 4 and the appendices.

---

> ### Author Response · Authors · 2025-11-19
> **Response 2 to Question 1: Span Length Influence**
>
> > **Q1: How sensitive is performance to span length? Is there any correlation between span length and policy learning effectiveness?**
>
> **A1:**
>
> Thank you for this insightful question. Our method does not impose fixed span lengths; spans are generated dynamically by the reward model based on response content and structure, as detailed in Appendix B. We address this concern by analyzing both the span length distribution and the relationship between span length and annotation quality.
>
> First, we examine how CoT reasoning affects span length selection. Table 5 shows the span length distribution statistics, comparing CoT-enabled and NoCoT variants on SLF5K. The results reveal that CoT reasoning produces slightly shorter Good spans but is more willing to precisely localize problematic regions in Poor spans. This suggests that CoT reasoning enables more targeted feedback by avoiding overly broad span selections. Q1, Q2, and Q3 represent the 25th-, 50th-, and 75th-percentile average token lengths per span, respectively.
>
> ---
>
> **Table 5: Span Length Distribution Statistics (SLF5K)**
> | Metric | Good Spans (CoT) | Good Spans (NoCoT) | Poor Spans (CoT) | Poor Spans (NoCoT) |
> | -------- | ---------------- | ------------------ | ---------------- | ------------------ |
> | Mean | 8.59 | 8.09 | 6.94 | 7.28 |
> | Median | 7.5 | 7 | 6 | 7 |
> | Min | 2 | 2 | 1 | 1 |
> | Max | 27 | 29 | 22 | 18 |
> | Q1 (25%) | 5 | 5 | 4 | 5 |
> | Q2 (50%) | 7.5 | 7 | 6 | 7 |
> | Q3 (75%) | 11 | 10 | 9 | 9 |
>
> ---
>
> Then, we quantify whether span length affects annotation quality by measuring accuracy across different length buckets. Table 6 shows span annotation accuracy stratified by quartiles of span length. Critically, with CoT reasoning, annotation accuracy remains consistently high (93.1%–96.4%) across all length ranges for both Good and Poor spans, demonstrating robustness to span length variation. In contrast, without CoT, we observe boundary effects: accuracy is notably higher in the Q1–Q2 range (94.1%–94.9%) but drops in shorter (0–Q1) and longer (Q4–100%) spans, suggesting that NoCoT struggles with both very short and very long span selections.
>
> **Table 6: Span Annotation Accuracy by Length Quartile (SLF5K)**
> | Length Range | Good Spans (NoCoT) | Good Spans (CoT) | Poor Spans (NoCoT) | Poor Spans (CoT) |
> | ------------ | ------------------ | ---------------- | ------------------ | ---------------- |
> | 0–Q1 | 89.80% | 96.10% | 93.30% | 96.40% |
> | Q1–Q2 | 94.10% | 93.40% | 94.90% | 96.20% |
> | Q2–Q3 | 90.50% | 95.60% | 92.00% | 94.70% |
> | Q3–100% | 91.20% | 93.10% | 91.40% | 95.30% |
>
> Length ranges defined by quartiles from Table 5. Accuracy measured as agreement with human-annotated ground truth.
>
> ---
>
> This length-invariant accuracy with CoT reasoning directly improves downstream performance.  As shown in Table 2 in our paper, Text2Grad with CoT achieves ROUGE-L 0.291 and BERTScore 0.902, outperforming the NoCoT variant (0.275 and 0.898, respectively). This performance gap confirms that inaccurate or noisy span selections—particularly the boundary effects observed without CoT—degrade policy learning effectiveness.
>
> CoT reasoning enables dynamic, content-driven span selection that maintains 93–96% annotation accuracy across all lengths, providing clean signals for policy gradients regardless of granularity. This robustness is essential for effective learning, as the performance improvements demonstrate.

---

> ### Author Response · Authors · 2025-11-19
> **Response 3 to Question 3 & 4: PRM Details and Recall Influence**
>
> > **Q3: The PRM-PPO results are reported on SLF5K and KodCode but not on UltraFeedback due to annotation cost. Could you clarify (a) how PRM spans or steps were defined in each domain, and (b) what cost differences led to excluding PRM-PPO on UltraFeedback, given that GPT-4o based step-level annotation might also be applicable?**
>
> **A3:**
>
> Thank you for the question regarding the PRM-PPO setup and its exclusion on UltraFeedback.
>
> **(a) Definition of PRM spans or steps.**
>
> For a fair comparison, we followed the methodology of Lightman et al. (2023, ICLR), which defines PRMs through explicit step-level supervision. On SLF5K (summarization), GPT-4o decomposed each response into sentence- or clause-level content units, assigning binary correctness labels based on factual alignment with the reference summary. On KodCode (code generation), GPT-4o segmented each program into code blocks or logical statements and labeled each according to unit-test outcomes or reference execution traces. These labeled spans served as the oracle for reward-model training and subsequent PPO optimization.
>
> **Table 7. Step-Level Annotations and PRM F1 Scores for SLF5K and KodCode**
> | Dataset | PRM F1 (%) | Annotation Note |
> | ------- | ---------- | ------------------------------------------------------------------------------------------- |
> | SLF5K | 78.17 | Sentence-/clause-level factual correctness based on alignment with the reference summary |
> | KodCode | 80.63 | logical-statement correctness based on unit-test and execution-trace outcomes |
>
> **(b) Exclusion on UltraFeedback.**
>
> As detailed in Section 4.5, we excluded PRM-PPO from UltraFeedback for principled reasons. UltraFeedback is an open-domain QA task with relatively long responses (averaging ~276 tokens) that lack explicit intermediate reasoning steps. Prior work demonstrates that PRMs are most effective for tasks with structured, multi-step reasoning such as mathematics and code generation where intermediate verification is natural and verifiable (Lightman et al., 2023; Zheng et al., 2025).
>
> Applying PRM supervision to UltraFeedback would require artificially decomposing open-ended answers into reasoning steps and verifying each one which is a process that lacks clear grounding in this domain and would incur substantial annotation costs without corresponding benefits. Outcome-level feedback remains the appropriate evaluation signal for general QA tasks.
>
> **References:**
>
> [1] Lightman, H., Kosaraju, V., Burda, Y., et al. Let’s Verify Step by Step. ICLR 2023.
>
> [2] Zheng, C., et al. A Survey of Process Reward Models: From Outcome Signals to Process Supervisions for Large Language Models. arXiv:2510.08049, 2025.
>
> > **Q4: Table 1 shows your reward model has moderately negative token recall (22% and 43% on UltraFeedback and SLF5K respectively). How sensitive is Text2Grad to reward model quality?**
>
> **A4:**
>
> Thank you for this important question about moderate negative-token recall and its impact on Text2Grad's sensitivity.
>
> The moderate recall values reflect label imbalance in our datasets. Approximately negative tokens represent just 11.4% of all tokens in SLF5K and 15.2% in UltraFeedback. Higher recall typically comes from mislabeling neutral tokens as positive or negative, introducing false positives that directly corrupt policy gradients.
>
> From a policy-learning perspective, precision is more critical than recall: a single false reward propagates through the NL-Gradient update and misleads optimization, while missing correct tokens merely reduces update density without harming correctness.
> Empirically, CoT-conditioned reward model achieves higher precision and stronger human alignment. The moderate recall reflects a deliberate precision-first design that favors selective, high-confidence supervision over dense but noisy labeling.
>
> **Table 8. Reward Model Quality and Policy Performance Sensitivity**
> | Dataset | RM Configuration | -1 Precision | -1 Recall | Policy Performance |
> | ----------------- | ---------------- | ------------ | ------------- | ---------------------- |
> | **SLF5K** | With CoT | 0.58 | 0.43 | Baseline |
> | | Without CoT | 0.53 (–5 pp) | 0.40 (–3 pp) | –1.6% / –21% / –1.2% |
> | **UltraFeedback** | With CoT | 0.46 | 0.22 | Baseline |
> | | Without CoT | 0.40 (–6 pp) | 0.35 (+13 pp) | –5.8% / –10.6% / –0.4% |
>
> *Note: "pp" denotes percentage points. Performance reports relative changes in ROUGE-L / BLEU / BERTScore F1 (SLF5K), and AlpacaEval 2.0 / ARC-c / MT-Bench (UltraFeedback).*
>
> Table 8 demonstrates that this precision first design translates to policy improvements. Despite moderate recall, CoT-enhanced reward modeling produces consistent gains across both datasets. Notably, when we ablate CoT rationales, UltraFeedback's recall increases by 13 percentage points but precision drops by 6 points, resulting in degraded policy performance. This confirms that Text2Grad's performance depends primarily more on reward precision and human alignment.

---

> ### Author Response · Authors · 2025-11-27
> **Gentle Check on Any Remaining Concerns Before the Discussion Deadline**
>
> Dear Reviewer D36W,
>
> As the discussion period is nearing its end, we would like to briefly confirm whether our previous responses have addressed your main concerns, particularly:
>
> * the new **analyses and ablations on reward granularity and token-level weighting**,
> * the added **measurements of training and annotation overhead**, and
> * the additional experiments on **span length, PRM-PPO setup, and reward model quality** (Q1–Q4).
>
> We truly appreciate your thoughtful review and the time you have dedicated to our work, and we are very glad to have had the opportunity to respond to your questions and provide further clarifications.
>
> Best regards,
> Authors

---

### Official Review · Reviewer_mXTR · 2025-11-07

**Soundness:** 3
**Presentation:** 3
**Contribution:** 2
**Rating:** 4
**Confidence:** 3

**Summary:**

This paper proposes a reinforcement learning (RL) method called TEXT2GRAD, which turns textual feedback into span-level gradients. This method contains three core components, including: (1) a feedback annotation pipeline that pairs critiques with token spans; (2) a reward model that predicts span-level rewards on answers while generating explanatory critiques; and (3) a span-level policy optimizer that back-propagates natural-language gradients. Experimental results on three tasks demonstrate that natural language feedback, when converted to gradients, is powerful for policy optimization.

**Strengths:**

1. The motivation of the proposed method is intuitive and the proposed method can support the motivation.
2. The empirical performance is validated through three different tasks, which shows the generalization ability of the proposed method.
3. This paper is overall well-written and easy to follow.

**Weaknesses:**

1. This paper misses existing works on reinforcement learning from natural language feedback especially in the training process, such as [1]. Since the proposed method exactly falls into this line of work, the authors should add a discussion to highlight the core novelty of the proposed method.

2. The proposed method jointly generates free-form natural language critiques and structured span-level reward labels. But how natural language critiques can help improve the learning of span-level rewards is not clear. From Figure 1, the relationship between these two modules seems weak, which should be further clarified.

3. The experimental part still has a lot of room for improvement. Now this part only include the main results of three datasets and a case study. In my view, at least a detailed ablation study should be conducted to show the contribution of the core parts in the proposed method.


[1] Learning from Natural Language Feedback. TMLR 2024.

**Questions:**

I have included my questions in the weaknesses part.

---

> ### Author Response · Authors · 2025-11-19
> **Response to Weaknesses 1–2: Method Definition & Critique–Span Coupling (1)**
>
> We are grateful for your constructive feedback. Below, we respond to each of your comments in detail.
> >  **W1: This paper misses existing works on reinforcement learning from natural language feedback, especially in the training process, such as [1]. Since the proposed method exactly falls into this line of work, the authors should add a discussion to highlight the core novelty of the proposed method.**
>
> **Response:**
> Thank you for the pointer. Conceptually, that work uses free-form feedback to produce refined sequences and then imitation learns on those refinements under a supervised/KL view; the learning signal remains sequence-level targets derived from edited outputs. By contrast, our method uses critiques to create span-aligned supervision that drives token-level policy optimization. Our reward model jointly generates critiques and span annotations, which we convert to token-level advantages for gradient-based updates via the NL-Gradient policy update. This yields localized credit assignment and interpretable updates tied to critique, rather than imitating edited sequences. We will cite **Learning from Natural Language Feedback** and highlight our core novelties.
>
> ---
>
> > **W2: The proposed method jointly generates free-form natural language critiques and structured span-level reward labels. But how natural language critiques can help improve the learning of span-level rewards is not clear. From Figure 1, the relationship between these two modules seems weak, which should be further clarified.**
>
> **Response:**
> Thank you for raising this question. The connection between critiques and span-level rewards is not weak; our method is explicitly designed so that critiques drive the span annotations. As described in Sections 3.3–3.4, the reward model generates the critique and the span labels jointly in a single autoregressive sequence, not as independent components. The annotation prompt (Appendix B) enforces a fixed ordering in which GPT-4o must first articulate the reasoning and only then provide spans that are verbatim excerpts justified by that reasoning. This creates a deterministic causal chain from critique → span selection → token-level pseudo-rewards.
>
> To make this coupling concrete, we summarize the human-alignment results from Appendices M and N, together with the span-quality win-rates comparing *w/ CoT* versus *w/o CoT* spans in Table 1 of our paper:
>
> ---
>
> ### **Table 1. Critique–Span Alignment Quality (Appendices M & N) and CoT vs. no-CoT Span Win-Rates**
>
> | Dataset       | Human Accuracy | Unmatched Ratio | Win-Rate (W:T:L) |
> | ------------- | -------------- | --------------- | ---------------- |
> | SLF5K         | 86%            | 1.30%           | 62 : 9 : 29  |
> | UltraFeedback | 82%            | 2.47%           | 53 : 9 : 38  |
> | KodCode       | 94%            | 0.93%           | 72 : 7 : 21  |
>
> **Human Accuracy**: Expert majority-vote agreement on whether (1) critiques correctly identify strengths/weaknesses and (2) spans are properly justified.
>
> **Unmatched Ratio**: Percentage of predicted spans that cannot be matched to the response text (Appendix M), indicating hallucinated or ungrounded labels.
>
> **Win-Rate**: LLM-as-judge preference between *CoT-conditioned spans* and *non-CoT spans*; CoT consistently yields higher-quality span annotations.
>
> ---
>
> Empirically, these results indicate a strong and reliable linkage: **82–94% of critique–span pairs are rated as consistent** across datasets, and **fewer than 2.5% spans** fail grounding checks, confirming that spans correspond to actual quoted regions rather than paraphrases. These alignment metrics indicate that our span-level supervision is reliably grounded in the underlying response text and not based on hallucinated reasoning.
>
> In addition, the Win-Rate results originally reported in Table 1 above come from an LLM-as-Judge comparison between *w/ CoT* and *w/o CoT* span annotations. Across all three datasets, **CoT-conditioned spans are consistently preferred**, indicating higher annotation quality after getting language feedback by reasoning.
>
> Furthermore, our ablations in Tables 2, 3, and 4 of our paper demonstrate the same trend at the policy-learning level: the *w/o CoT* variant (“No natural feedback”) performs worse than the full TEXT2GRAD pipeline. When the model **first generates a critique and then produces spans conditioned on that reasoning**, the resulting span-level rewards yield systematically stronger downstream policy performance.
>
> Natural-language critiques guide span-level rewards by first explaining why a segment is right or wrong, which then anchors where the reward should be applied; this reasoning-before-labeling step consistently produces more precise, grounded, and informative spans. We will further clarify the causal relationship between critiques and span-level reward generation in the revision.

---

> ### Author Response · Authors · 2025-11-19
> **Response to Weaknesses 3: Experimental Analysis and Ablations (2)**
>
> > **W3: The experimental part still has a lot of room for improvement. Now this part only includes the main results of three datasets and a case study. In my view, at least a detailed ablation study should be conducted to show the contribution of the core parts in the proposed method.**
>
> **Response:**
>
> Thank you for pointing this out. We agree that the contribution of each component should be made explicit. In our revision, we reorganize existing ablations from the main paper to clearly highlight the role of CoT critiques and introduce new ablations on reward granularity, feedback source, and within-span token weighting strategies.
>
> **i. (Reorganized from main paper) Effect of CoT Critiques on Span-Level Rewards**
>
> We vary only whether the reward model is prompted to generate a CoT critique before producing span labels. Without CoT, the model directly outputs span-level signals. With CoT, the critique is generated first and conditions the span map, which is then projected into token-level pseudo-rewards for NL-Gradient Update. CoT reasoning consistently improves performance by sharpening the justification used during span selection in the same decoding pass, yielding cleaner span maps and more informative token-level pseudo-rewards for NL-Gradient Update.
>
> **Table 2.1: Ablation of CoT Critiques — KodCode**
> | Method (8B) | HumanEval Base | HumanEval Plus | MBPP Base | MBPP Plus | Avg. |
> | ------------------- | -------------- | -------------- | --------- | --------- | ---- |
> | Text2Grad (w/o CoT) | 63.8 | 57.3 | 62.2 | 53.4 | 59.2 |
> | Text2Grad (w/ CoT) | 67.7 | 61.6 | 73.3 | 61.6 | 66.1 |
>
> **Table 2.2: Ablation of CoT Critiques — UltraFeedback**
> | Method | AlpacaEval 2.0 | ARC-C | MT-Bench |
> | ------------------- | -------------- | ----- | -------- |
> | Text2Grad (w/o CoT) | 28.6 | 83.1 | 7.49 |
> | Text2Grad (w/ CoT) | 34.7 | 84.4 | 7.58 |
>
> **Table 2.3: Ablation of CoT Critiques — SLF5k**
> | Method | ROUGE-L | BERTScore F1 |
> | ------------------- | ------- | ------------ |
> | Text2Grad (w/o CoT) | 0.275 | 0.898 |
> | Text2Grad (w/ CoT) | 0.291 | 0.902 |
>
> **ii. (Newly added) Effect of Reward Granularity and Feedback Source**
>
> To further isolate the contribution of span-based reward construction, we conduct new ablations on SLF5K that examine three orthogonal dimensions: supervision granularity (dense token-level vs. span-level), feedback source (human vs. model-generated), and within-span token weighting strategies.
>
> We compare five configurations. Dense token labeling provides maximal supervision density, with approximately 70% of tokens receiving labels (mostly verbs and function words), but it produces noisy advantages and unstable policy updates while requiring 6–8 times more annotation tokens. Span-level mapping with human feedback performs better, confirming the structural value of span selection over dense labeling.
>
> For within-span token weighting, we test two alternative strategies against our uniform assignment. Token Importance assigns full rewards (±1.0) only to nouns and verbs within each span, while assigning reduced weights (±0.2) to others, under the hypothesis that content words carry greater semantic responsibility. Linear Decay assigns rewards proportionally to token position: for a span of length $n$, the $i$-th token receives $r_i = 1.0 - (i-1) \cdot \frac{0.9}{n-1}$ for positive spans (and negated for negative spans), concentrating credit on early tokens under the assumption that initial errors dominate in autoregressive generation.
>
> As shown in Table 3, both alternative weighting schemes underperform uniform assignment. Token Importance introduces part-of-speech classification noise that disrupts span coherence, while Linear Decay amplifies variance in the advantage estimator by concentrating gradients on initial tokens. The full Text2Grad pipeline with uniform weighting achieves the best results, demonstrating that balanced credit assignment combined with structured CoT reasoning outperforms dense labels, human annotation, and gradient reweighting heuristics.
>
> **Table 3: Ablation of Reward Granularity, Feedback Source, and Token Weighting — SLF5K**
>
> | Model                                    | ROUGE-L | BLEU  | BERTScore F1 |
> |------------------------------------------|---------|-------|--------------|
> | Dense Token Reward                       | 0.196   | 0.022 | 0.888        |
> | Text2Grad (Token Importance)             | 0.237   | 0.059 | 0.889        |
> | Text2Grad (Linear Decay)                 | 0.253   | 0.068 | 0.890        |
> | Text2Grad (Human Feedback, Uniform)      | 0.286   | 0.091 | 0.900        |
> | **Text2Grad (GPT-4o + CoT, Uniform)**        | **0.291**   | **0.094** | **0.902**        |
>
> We will incorporate these discussions and additional ablations in our revision. Thank you again for your valuable suggestions! If you have any further questions, please feel free to ask, and we are happy to address them.

---

> ### Author Response · Authors · 2025-11-27
> **Gentle Check on Any Remaining Concerns Before the Discussion Deadline**
>
> Dear Reviewer mXTR,
>
> As the discussion period is drawing to a close, we would like to gently check whether our previous responses have addressed your main concerns, particularly:
>
> * the added discussion of **prior work on learning from natural language feedback** (W1), and
> * the new **analyses and ablations** clarifying the coupling between critiques and span-level rewards, as well as the experimental contributions (W2–W3).
>
> We would be very grateful if you could let us know whether any remaining questions or points would benefit from further clarification, especially while there is still time to run additional experiments if needed. Thank you again for your thoughtful review and the time you have devoted to our work!
>
> Best regards,
> Authors

---

### Author Response · Authors · 2025-11-25
**Summary of Key Review Points and Our Responses**

Dear PCs, ACs, and Reviewers,

We sincerely thank all committees for your tremendous efforts devoted to this paper under such an unexpected accident. We truly appreciate the professionalism, patience, and extra workload undertaken by the ACs and PCs in organizing and safeguarding the review process. We are also deeply grateful for the constructive, careful, and high-quality comments provided by all reviewers.

To ease the burden on the AC, we objectively and concisely summarize the major common review concerns and our rebuttal progress before the information leakage incident.

- ***Feedback and Annotation Quality:*** We clarified the LLM-based annotation pipeline (Appendix B) and human evaluation setup (Appendices M & N). Three expert annotators exhibit **82–94% agreement** with GPT-4o critiques and spans, and the **unmatched-span ratio is below 2.5%** across datasets. Our structured prompts enforce a *critique to span* justification pattern, and we adopt a **precision-first reward model design**: moderate recall mainly reduces update density, whereas high precision is crucial for stable and beneficial policy updates. Together, these support a high-quality annotation pipeline with precise, well-grounded span annotations.

---

- ***Reward design, CoT ablations, and span robustness:***  We conducted ablations comparing dense token rewards and within-span weighting schemes against uniform span rewards, and found that **uniform span-level rewards consistently outperform both denser and more “fine-grained” alternatives**, despite using fewer labeled tokens and lower annotation cost. We also clarified the role of CoT: CoT-based span generation yields higher span accuracy and consistently better policy performance. Importantly, our span-length analysis shows that with CoT, span annotation accuracy remains **consistently high (≈93–96%) across all length quartiles**, whereas the no-CoT variant degrades on very short and very long spans. This indicates that CoT produces **length-robust span supervision** that remains reliable from short phrase-level spans to longer clause-level regions, providing clean, well-localized token-level rewards and explaining why Text2Grad configuration is the most effective and stable among all tested designs.

---

- ***Baselines and feedback sources:*** We added **ILF** as a strong *SFT-with-feedback* baseline and found that Text2Grad consistently outperforms both PPO and ILF across all three benchmarks. We further showed that Text2Grad trained on **human-written critiques** still significantly improves over the base model, indicating **robustness to feedback source** and realistic noise. We also clarified our **PRM-PPO** setup and reported PRM accuracies for SLF5K and KodCode.

---

- ***Generality, overhead, and robustness:*** To assess generality, we added experiments on **Mistral-7B-Instruct**, showing that Text2Grad transfer across **model families & scales**. We provided **training time** and **annotation-cost** measurements, showing only modest overhead: one extra reward-model forward pass per trajectory and a per-sample annotation cost on the order of **10⁻³ USD**. We also clarified that spans are first matched at the **character level** and then mapped to the policy model’s tokenizer, making reward attribution **robust to tokenizer differences**.

---

- ***Terminology and presentation:*** We clarified that **“NL-Gradient”** refers to policy gradients **weighted by feedback-derived token-level rewards**, not differentiation through language itself. We also fixed minor issues (notation, typos, figure/text mismatches) and expanded related work to better position Text2Grad relative to span-level RLHF and process supervision methods, such as *MA-RLHF*, *SCAR*, and language-based reward shaping.

---

- ***Stability and human evaluation:*** Beyond automatic metrics, we additionally evaluate stability and human-perceived gains. We report low variance across perturbed inference hyperparameters and show that human annotators consistently prefer Text2Grad over PPO/DPO outputs in side-by-side evaluations, confirming that the improvements are robust and not specific to a single judge.

---

We again thank the reviewers and the AC for your thoughtful feedback, and hope that these clarifications, analyses, and new experiments will be helpful for your further assessment of the paper!

Best regards,
Text2Grad Team

---

### Meta-Review · Area_Chair_SQLQ · 2026-01-03

**Summary:**

The rebuttal addressed most of the main reviewer concerns.

**Reviewer Concerns:**

The rebuttal clarified novelty and mentioned adding more related work, added more baselines and provided ablation studies, added human agreement checks. On the other hand, concerns about statistical significance checks, broader human evaluation, and different noise setups remain, but these can be added in the camera ready, if possible.

**Reviewer Scores:**

The initial ratings were 46266. The two negative reviewers' concerns were mostly addressed, and it's possible that their ratings might have increased if we had a full discussion period.

---

### Decision · Program_Chairs · 2026-01-26

Accept (Poster)